# Fast Text-to-Audio Generation with One-Step Sampling via Energy-Scoring and Auxiliary Contextual Representation Distillation

## Abstract

Autoregressive (AR) models with diffusion heads have recently achieved strong text-to-audio performance, yet their iterative decoding and multi-step sampling process introduce high-latency issues. To address this bottleneck, we propose a one-step sampling framework that combines an energy-distance training objective with representation-level distillation. An energy-scoring head maps Gaussian noise directly to audio latents in one step, eliminating the need for a costly recursive diffusion sampling process, while distillation from a masked autoregressive (MAR) text-to-audio model preserves the strong conditioning learned during diffusion training. On the AudioCaps benchmark, our method consistently outperforms prior one-step baselines on both objective and subjective metrics while substantially narrowing the quality gap to AR diffusion systems with multi-step sampling. Compared to the state-of-the-art AR diffusion system, IMPACT, our approach achieves up to 25× faster inference with highly competitive audio quality. These results demonstrate that combining energy-distance training with representation-level distillation provides an effective recipe for fast, high-quality text-to-audio synthesis.

## 1 Introduction

With the rapid growth of user-generated content, personalized audio generation has become increasingly important. Recent advances in text-to-audio (TTA) generation aim to synthesize audio directly from natural language prompts, allowing humans to engage with the models more intuitively and with less technical effort. Driven by advances in deep generative models, TTA generation has made significant progress. Nowadays, latent diffusion models (LDMs; Rombach et al., 2022) have become a leading approach, achieving state-of-the-art results on challenging TTA benchmarks such as AudioCaps (Kim et al., 2019).

Autoregressive continuous sampling (Li et al., 2024) is a recent trend in generative models that combines the power of autoregressive (AR) transformers with a sampling method such as diffusion (Ho et al., 2020) or flow matching (Lipman et al., 2023), to generate continuous latents. This approach is highly effective because it avoids the information loss problem often seen in discrete-based models, while enabling models to generate content progressively. Instead of producing an entire sequence at once, the model incrementally generates content, using prior outputs as context for subsequent iterations. The iterative process of modeling continuous latents leads to high-quality results and has shown strong performance in many different modalities, including image (Li et al., 2024; Fan et al., 2025), video (Zhang et al., 2025), speech (Jia et al., 2025), audio (Yang et al., 2025; Huang et al., 2025), and multi-

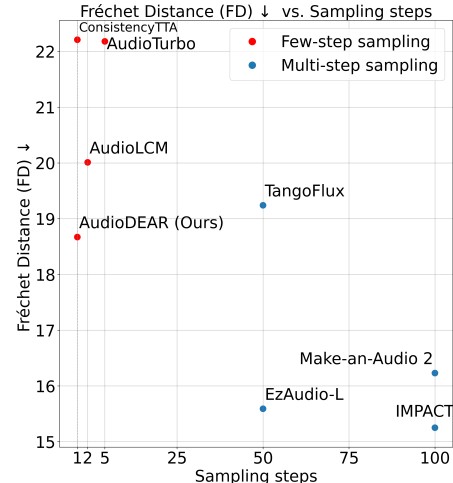

Figure 1: FD vs sampling steps.

modal large language models (Sun et al., 2024b). However, despite its strong quality of generation, the inherent iterative nature of both the autoregressive decoding and the diffusion sampling process contributes to considerable inference latency, which presents a critical trade-off between generation quality and inference speed, making them impractical for real-time interactive applications.

In this context, a key challenge is the computational cost of the generative process. For an AR model with autoregressive decoding of $r$ iterations, and each decoding iteration requiring an $n$-step sampling process, such as diffusion (Ho et al., 2020) or flow matching (Lipman et al., 2023), the total generation process requires $r \times n$ sampling steps, leading to significant inference latency. A natural way to accelerate generation is to reduce the steps of either $r$ or $n$. However, prior work, including Diffsound (Yang et al., 2023), SoundStorm (Borsos et al., 2023), MAGNET (Ziv et al., 2024), MaskGIT (Chang et al., 2022), MAR (Li et al., 2024), and IMPACT (Huang et al., 2025), indicates that overly aggressive reduction of AR steps ($r$) leads to substantial generative quality degradation. Therefore, reducing the number of sampling steps ($n$) is a more suitable strategy to achieve faster generation with minimal compromise to output quality. To accelerate this process, recent research (Song et al., 2021; Karras et al., 2022; Lu et al., 2022; 2025; Zheng et al., 2023; Liu et al., 2022; Bao et al., 2022; Zhang & Chen, 2022; Song et al., 2023; Salimans & Ho, 2022; Frans et al., 2025; Geng et al., 2025) focuses on reducing the number of sampling steps for generation. However, a consistent limitation is that the quality of one-step sampling ($n = 1$) for generation remains inferior to that of multi-step sampling. For example, in the image generation field, Shortcut Model (Frans et al., 2025) reduces sampling steps and improves over naive flow matching, but their one-step sampling quality still lags behind multi-step approaches. Similarly, MeanFlow (Geng et al., 2025) outperforms prior one-step diffusion and flow matching models, yet struggles to generate high-quality outputs under small model configurations autoregressively. In the results, we demonstrate that both Shortcut and MeanFlow training objectives are ineffective under AR sampling frameworks for TTA generation.

To address the downside of existing AR diffusion approaches that require multiple sampling steps, we propose AUDIODEAR, a **D**istillation-enhanced **E**nergy-scoring **A**uto**R**egressive model for text-to-**Audio** generation, which integrates an energy-based training objective (Székely & Rizzo, 2013) with distillation techniques to achieve fast and high-quality audio generation. Building on current state-of-the-art diffusion-based models (Huang et al., 2025) for TTA generation, we replace the diffusion loss with an energy-distance objective, a statistical estimate that measures the discrepancy between two probability distributions based on expected pairwise distances between samples. The reformulated training objective allows the model to learn to map raw noise vectors directly to audio latents, removing the need for multiple sampling steps ($n$). To further close up the performance gap between our one-step[1] generation method and multi-step generation models, we further adopted an additional distillation loss between the transformer backbones of a diffusion-based variant and our proposed energy-scoring framework. Introducing this auxiliary distillation loss into the training objective led to consistent improvements across all objective metrics, including Fréchet Distance (FD; Heusel et al., 2017), Fréchet Audio Distance (FAD; Kilgour et al., 2018), Kullback–Leibler divergence (KL), Inception Score (IS; Salimans et al., 2016), and Contrastive Language-Audio Pre-training (CLAP; Wu et al., 2023) score. Overall, our AUDIODEAR outperforms existing fast consistency-based (Song et al., 2023) TTA generation models targeting few-step sampling, such as ConsistencyTTA (Bai et al., 2023), AudioLCM (Liu et al., 2024b), and AudioTurbo (Zhao et al., 2025), on both objective and subjective metrics, while narrowing the performance gap between one-step and multi-step sampling approaches. In summary, our contributions of this work are:

- We are the first to apply the energy-distance objective in TTA generation, enabling one-step latent synthesis with low latency.

- We leverage a diffusion-based transformer backbone as a fixed teacher, and introduce an auxiliary distillation loss that aligns its feature representations with those of our energy-based model, yielding consistent improvements across all objective performance metrics on the AudioCaps benchmark.

- We surpass ConsistencyTTA, AudioTurbo, and AudioLCM on FD score under a one-step sampling budget constraint as shown in Figure 1, as well as KL, IS, and CLAP score.

---

[1]The term "one-step" refers to one sampling step with the sampling module. The model still requires $r$ autoregressive iterations for generation.

## 2 RELATED WORK

### 2.1 AUTOREGRESSIVE MODELS WITH SAMPLING HEAD

A significant trend in generative modeling is the integration of autoregressive (AR) models with sampling heads to handle continuous data modalities, thereby avoiding the information loss associated with traditional vector quantization shown in existing work (Yuan et al., 2024; Xu et al., 2024; Fan et al., 2025). The autoregressive nature of these models is crucial, as it allows further iterations to utilize previously generated content as context, progressively generating the output, and enhancing predictive capabilities in subsequent steps. Pioneering this approach, Li et al. (2024) introduced the masked autoregressive (MAR) (Li et al., 2024) model, which adopts a diffusion loss in place of the standard cross-entropy loss. In this framework, the AR model predicts a conditioning vector for each position of a sequence, which then guides a lightweight diffusion head to generate the continuous-valued latents. This core framework was successfully scaled for text-to-image generation in Fluid (Fan et al., 2025) and adapted for efficient TTA synthesis in IMPACT (Huang et al., 2025). This paradigm has also been adapted by several LLM-style, decoder-only transformers for various applications and demonstrated huge success in applications like multimodal generation and understanding (Sun et al., 2024b), image generation (Gao & Shou, 2025), video generation (Zhang et al., 2025), speech generation (Jia et al., 2025), and spoken chatbots (Zeng et al., 2024). Though performing well across various tasks, the main problem of this AR sampling framework is the inference speed, as each AR step requires a large number of sampling steps for generation.

### 2.2 FEW-STEP SAMPLING

Diffusion models deliver high-fidelity outputs but incur significant inference cost. Training-free samplers such as DDIM (Song et al., 2021), Heun (Karras et al., 2022), the DPM-Solver family (Lu et al., 2022; 2025; Zheng et al., 2023), PNDM (Liu et al., 2022), Analytic-DPM (Bao et al., 2022), and DEIS (Zhang & Chen, 2022) can reduce the number of sampling steps to the order of tens, yet struggle to push below 10 steps for generation tasks. Recent breakthrough methods like Shortcut models (Frans et al., 2025) and MeanFlow (Geng et al., 2025) have achieved significant progress in few-step image generation with under 4 steps, yet substantial quality gaps persist between these fast approaches and their multi-step counterparts. This performance disparity is particularly pronounced when using smaller model configurations commonly employed in research settings with less than 200M parameters, where the trade-off between latency and generation quality remains a key challenge. In this work, we address the problem of few-step sampling through energy-scoring models, which only requires one step for sampling, while maintaining good quality and semantic relevance for TTA generation. In Appendix G, we demonstrate visualization results of a toy dataset of different continuous sampling strategies, showcasing the limitations of existing few-step sampling methods.

### 2.3 GENERATIVE MODELS WITH ENERGY-DISTANCE SCORING

Energy-scoring methods (Székely, 2003) generate samples in one forward pass by minimizing a distance-based scoring rule, enabling fast sampling, whereas diffusion (Ho et al., 2020) and flow matching (Lipman et al., 2023) methods require solving iterative denoising or flow steps, often tens to hundreds, making generation much slower. Building on these advantages, energy-distance training objectives have been applied in generative modeling for various tasks, including image generation (Bellemare et al., 2017; Shao et al., 2025), text-to-speech (Gritsenko et al., 2020), and time series modeling (Pacchiardi & Dutta, 2022; Pacchiardi et al., 2024). However, the use of these objectives for sound event audio generation remains a relatively unexplored area. In this work, we demonstrate that energy-scoring effectively accelerates TTA generation and can be further enhanced with representation distillation to deliver high-quality, high-fidelity, and high-text-relevance audio.

## 3 METHOD

### 3.1 BACKGROUND: MASKED AUTOREGRESSIVE CONTINUOUS SAMPLING

Masked autoregressive (MAR) continuous sampling (Li et al., 2024) conducts the autoregressive modeling paradigm in continuous latent spaces. In contrast to discrete token prediction, this frame-

work generates high-dimensional latent variables at each iteration through a continuous sampling head, thereby alleviating the information loss typically encountered in discrete tokens.

Training is carried out under a masked generative modeling framework. Given a latent sequence $y = \{y^1, \cdots, y^L\} \in \mathbb{R}^{L \times d}$, where $d$ is the latent dimension, a random subset of positions is masked by replacing them with mask tokens. The partially masked latent sequence is then served as the input of the mask autoregressive transformer $\text{Enc}_\phi$ to generate a sequence of contextual representations $\{h^1, \cdots, h^L\} \in \mathbb{R}^{L \times D}$, where $D$ is the hidden dimension. For each masked position $i$, a continuous sampling head conditioned on $h^i$ predicts the masked latents, which are then compared against the ground truth latents at those corresponding positions. The training objective is defined with respect to the chosen sampling strategy, in our case, energy-scoring, which calculates the loss according to the energy-distance training objective elaborated in Section 3.2.

During inference, iterative parallel decoding (Chang et al., 2022) is adopted to generate latent sequences. This method generates an audio latent sequence through multiple decoding iterations, with each iteration generating a random subset of positions to gradually build up the whole sequence throughout the process. A major limitation of this approach is its reliance on multi-step sampling methods, such as diffusion (Ho et al., 2020) and flow matching (Lipman et al., 2023), as illustrated in Figure 2(c), to generate latents. This reliance substantially increases inference time. To address this limitation, we introduce a one-step sampling strategy based on energy scoring as illustrated in Figure 2(b), which requires only one forward pass and substantially reduces the latency of latent generation.

## 3.2 ENERGY-SCORING

Energy-scoring (Székely, 2003) provides a direct mapping from the source noise distribution to the latent space in one forward pass. The mapping is learned by optimizing the generated distribution of the model and that of the target distribution.

**Energy-distance.** Let $P$ and $Q$ be probability distributions on $\mathbb{R}^d$. According to (Székely, 2003), the *energy-distance* between $P$ and $Q$ is defined as

$$\mathcal{E}(P, Q) = 2 \mathbb{E}[\|X - Y\|] - \mathbb{E}[\|X - X'\|] - \mathbb{E}[\|Y - Y'\|], \tag{1}$$

where $X, X' \overset{\text{i.i.d.}}{\sim} P, \quad Y, Y' \overset{\text{i.i.d.}}{\sim} Q$, and $\|\cdot\|$ denotes the Euclidean norm (L2 norm) in $\mathbb{R}^d$. The energy-distance satisfies $\mathcal{E}(P, Q) \geq 0$, with equality if and only if $P = Q$ (See Appendix A for the proof). Larger values of $\mathcal{E}(P, Q)$ correspond to greater dissimilarity between the two distributions. In the context of model training, the random variables $X$ and $X'$ are drawn from the model's predictive distribution $P_\theta$ parameterized by $\theta$, while $Y$ and $Y'$ are drawn from the target distribution $Q$ corresponding to the ground truth training data. The term $\mathbb{E}[\|Y - Y'\|]$ depends only on $Q$ and is therefore *independent* of the model parameters $\theta$. Consequently, this term acts as an additive constant in the objective function and does not affect the optimization. By omitting the constant term $\mathbb{E}[\|Y - Y'\|]$, minimizing the energy-distance with respect to $\theta$ is therefore equivalent to minimizing

$$\widetilde{\mathcal{E}}(P_\theta, Q) = 2\mathbb{E}_{X \sim P_\theta, Y \sim Q}[\|X - Y\|] - \mathbb{E}_{X, X' \sim P_\theta}[\|X - X'\|]. \tag{2}$$

**Training objective.** During training, the expectations in Equation 2 can be estimated via Monte Carlo sampling. Specifically, for each data point $y$ drawn from $Q$, we draw two independent samples $x_1, x_2 \sim P_\theta$ from the model's predictive distribution, and compute the empirical estimate

$$\mathcal{L}_{\text{energy}} = \|x_1 - y\| + \|x_2 - y\| - \|x_1 - x_2\|. \tag{3}$$

Empirical justification for selecting two samples to estimate Equation 2 is provided in Section 5.5.

**Energy-scoring head.** As shown in Figure 2(a), in the training phase, when predicting the $i^{\text{th}}$ audio latent of a sequence, we first draw a noise vector $n_1 \sim \mathcal{N}(0, I)$. The energy-scoring head $F_\theta$ receives as input the contextual representation $h^i \in \mathbb{R}^D$ produced by the masked autoregressive transformer $\text{Enc}_\phi$ and the noise vector $n_1$, producing the first sample $x_1^i = F_\theta(h^i, n_1)$. Subsequently, an independent noise vector $n_2 \sim \mathcal{N}(0, I)$ is drawn, and the second sample is obtained analogously as $x_2^i = F_\theta(h^i, n_2)$. The two resulting samples $x_1^i$ and $x_2^i$ are then used to form the

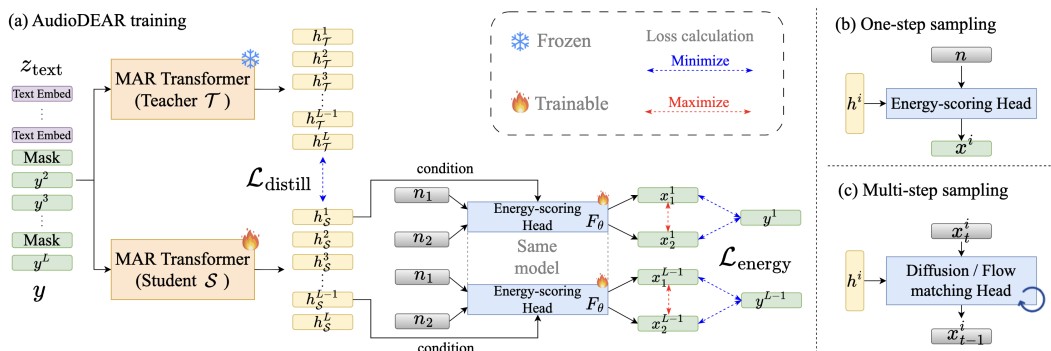

Figure 2: (a) Training pipeline of our energy-scoring framework with representation distillation. Positions 1 and $L-1$ are masked for demonstration. More details of the mask autoregressive sampling framework are described in Appendix D. (b) One-step sampling of our energy-scoring head at inference. The architecture of the energy-scoring head is elaborated in Appendix E. (c) Multi-step sampling of a diffusion/flow matching head at inference. Overall structural diagrams for training and inference are provided in Appendix I.

training objective in Equation 3, which minimizes the discrepancy between generated samples and the ground truth while maximizing the distance between different model samples.

As shown in Figure 2(b), in the inference phase, for each selected position for generation at each decoding iteration, the contextual representation $h^i$ and a Gaussian noise vector $n \sim \mathcal{N}(0, I)$ are provided to the energy-scoring sampling head to generate latent $x^i$. The main advantage of energy-scoring is that it generates latents in one forward pass, eliminating the need for multiple sampling steps, which are typically required for diffusion and flow matching.

**Classifier-Free Guidance in Representation Space** During inference, to achieve classifier-free guidance (CFG; Ho & Salimans 2022), we combine the text-conditioned representations with the null-conditioned representations with a CFG scaling value (Ma et al., 2025). More specifically, we compute two versions of this representation: a conditional one, $h^i_{\text{cond}}$, obtained by forwarding the text embeddings $z_{\text{text}}$ and audio latents into the transformer backbone $\text{Enc}_\phi$ and an unconditional one $h^i_{\text{uncond}}$, where the text pathway is replaced by null text embeddings $z_\emptyset$.

$$h^i = \text{CFG} \cdot h^i_{\text{cond}} + (1 - \text{CFG}) \cdot h^i_{\text{uncond}}, \tag{4}$$

where $h^i_{\text{cond}} = \text{Enc}_\phi(x, z_{\text{text}})^i$, $h^i_{\text{uncond}} = \text{Enc}_\phi(x, z_\emptyset)^i$, and $x$ denote the audio latent sequence generated during the inference phase. The advantage of performing CFG at the representation-level but not at the audio-latent-level is that this eliminates the need to forward the energy-scoring head $F_\theta$ twice to produce conditional and unconditional outputs.

## 3.3 REPRESENTATION DISTILLATION

To further bridge the performance gap between our proposed one-step energy-scoring model and multi-step diffusion counterparts, we introduce a *representation distillation* strategy from a strong teacher model as shown in Figure 2(a). Specifically, we employ the backbone transformer from IMPACT (Huang et al., 2025), trained with a diffusion loss, as the fixed teacher network. Our student network's backbone transformer shares the same architecture but is trained with the energy-distance objective described in Equation 3.

Let $\{h^1_{\mathcal{T}}, \cdots, h^L_{\mathcal{T}}\} \in \mathbb{R}^{L \times D}$ and $\{h^1_{\mathcal{S}}, \cdots, h^L_{\mathcal{S}}\} \in \mathbb{R}^{L \times D}$ denote the hidden representations at the final transformer block for the teacher ($\mathcal{T}$) and student ($\mathcal{S}$) models, respectively, where $L$ is the sequence length and $D$ is the hidden dimension. We align the final-layer representations of the student with those of the teacher by minimizing the mean squared error (MSE) between the corresponding hidden states:

$$\mathcal{L}_{\text{distill}} = \frac{1}{L} \sum_{i=1}^{L} \|h^i_{\mathcal{S}} - h^i_{\mathcal{T}}\|^2_2. \tag{5}$$

The final training objective for the student combines the energy-distance loss from Equation 3 with the distillation term:

$$\mathcal{L}_{\text{total}} = \mathcal{L}_{\text{energy}} + \lambda \cdot \mathcal{L}_{\text{distill}}, \tag{6}$$

where $\lambda$ is a hyperparameter controlling the influence of the distillation loss. By aligning the student's contextual representations with those of the teacher, we allow our energy-scoring framework to inherit the strong conditioning capabilities learned by the diffusion-trained transformer, without incurring the inference cost of multi-step sampling.

## 4 EXPERIMENTAL SETUP

### 4.1 DATASETS

We adopt the two widely used TTA datasets for training, AudioCaps (Kim et al., 2019) and Wav-Caps (Mei et al., 2024). Audio clips shorter than 10 seconds are zero-padded, while those exceeding 10 seconds are truncated by selecting a random contiguous 10-second segment. Following the Au-dioLDM (Liu et al., 2023) preprocessing protocol, each audio clip is standardized into a 10-second segment and transformed into a Mel spectrogram, resulting in approximately $1,200$ hours of audio. In addition, we sample 500 hours of audio from AudioSet (Gemmeke et al., 2017), resulting in a combined training corpus of 1700 hours. This dataset is used to train both the IMPACT teacher model[2] and our proposed energy-scoring model. More details on the training data set for each baseline model can be found in Appendix H.

For evaluation, we adopted the AudioCaps evaluation split, which consists of 964 audio samples, each paired with five textual descriptions. Following previous work (Liu et al., 2023; Hai et al., 2024; Huang et al., 2025), we randomly select one caption from each set as the conditioning text for TTA generation.

### 4.2 MODEL CONFIGURATIONS

The input audio is represented as a Mel spectrogram of size $(1024 \times 64)$, which is encoded into VAE latents of size $(256 \times 16 \times 8)$ using AudioLDM's VAE model. We adopt a patch size of $4$, flattening the patches into a sequence of length $256$ with a latent dimension $d$ of $128$. Textual information is incorporated by appending the Flan-T5 (Chung et al., 2024) and CLAP (Wu et al., 2023) text embeddings to the patched audio latents, following the IMPACT configuration, resulting in a text-embedding sequence of length $78$. For the transformer backbone, we employ the IMPACT-Base architecture, consisting of $24$ transformer layers with a hidden dimension $D$ of $768$. The energy-scoring head consists of residual MLP blocks, with the noise vectors provided as input to the energy-scoring head, while contextual representations $h^i$ are injected via adaptive normalization (Perez et al., 2018). Further details on the architecture of the energy-scoring head are provided in Appendix E. During training, we apply a masking rate randomly sampled from the range $[70, 100)$ to the audio latents, enabling masked generative modeling with the energy-distance objective. For representation distillation, we adopt the transformer backbone of the diffusion-based state-of-the-art model IMPACT (Huang et al., 2025) as the teacher, and integrate the distillation loss with the energy-distance objective using a distillation weight $\lambda = 1000$, as defined in Equation 6. Unless otherwise specified, we train with a batch size of $2048$ and a learning rate of $1e-3$. At inference time, we fix the number of decoding iterations to $64$, matching the default IMPACT configuration, and apply a classifier-free guidance scale of $4.0$.

### 4.3 EVALUATION

We evaluate our proposed TTA generation framework using both objective and subjective metrics. For objective assessment, we report Fréchet distance (FD; Heusel et al. 2017), Fréchet audio distance (FAD; Kilgour et al. 2018), Kullback–Leibler divergence (KL), and inception score (IS; Salimans et al. 2016) following the AudioLDM evaluation protocol [3], and CLAP similarity (Wu et al., 2023)

---

[2]Since IMPACT's official checkpoint was unavailable, we had to train the teacher model ourselves.

[3]https://github.com/haoheliu/audioldm_eval

Table 1: System-level performance of text-to-audio generation models. "Data" denotes the total training data duration of the model in hours, including the data involved during training the teacher model, if any. "Step" denotes the number of sampling steps required to sample an audio latent. "REL." and "OVL." denote the subjective evaluation reported as mean opinion score for text-relevance and overall audio quality, respectively. The subscripts denote the standard error. "Dist." stands for distillation. Detailed statistical measures for subjective evaluation are listed in Table 9 in Appendix F. Best performance values among the few-step sampling methods are marked in bold. Second-best performance values are marked with underlines.

| AudioCaps | Data | # para | Step | FD↓ | FAD↓ | KL↓ | IS↑ | CLAP↑ | REL↑ | OVL↑ |
|---|---|---|---|---|---|---|---|---|---|---|
| Ground Truth | - | - | - | - | - | - | - | 0.373 | $4.45_{\pm0.09}$ | $3.68_{\pm0.08}$ |
| **Discrete-based** | | | | | | | | | | |
| MAGNET-L | $\approx 4000$ | 1.5B | - | 26.19 | 2.36 | 1.64 | 9.10 | 0.253 | - | - |
| **Diffusion/Flow matching Models** | | | | | | | | | | |
| Tango 2 | $\approx 3333$ | 866M | 200 | 20.66 | 2.63 | 1.12 | 9.09 | 0.375 | $4.07_{\pm0.08}$ | $3.42_{\pm0.09}$ |
| TangoFlux | 3700 | 516M | 50 | 19.24 | 2.32 | 1.18 | 12.43 | 0.382 | - | - |
| EzAudio-L | >5500 | 596M | 50 | 15.59 | 2.25 | 1.38 | 11.35 | 0.391 | - | - |
| EzAudio-XL | >5500 | 874M | 50 | 14.98 | 3.01 | 1.29 | 11.38 | 0.387 | $4.03_{\pm0.08}$ | $3.31_{\pm0.07}$ |
| Make-an-Audio 2 | 3700 | 160M | 100 | 16.23 | 2.03 | 1.29 | 9.95 | 0.345 | - | - |
| AudioLDM2-full | 29510 | 346M | 200 | 32.14 | 2.17 | 1.62 | 6.92 | 0.273 | - | - |
| AudioLDM2-full-L | 1150k | 712M | 200 | 33.18 | 2.12 | 1.54 | 8.29 | 0.281 | - | - |
| AudioMNTP | 1200 | 193M | 100 | 14.81 | 1.68 | 1.16 | 9.67 | 0.336 | - | - |
| IMPACT | 1700 | 193M | 100 | 15.25 | 1.26 | 1.06 | 10.57 | 0.372 | $4.38_{\pm0.10}$ | $3.47_{\pm0.09}$ |
| **Few-step Sampling** | | | | | | | | | | |
| ConsistencyTTA | 145 | 559M | 1 | 22.21 | 2.83 | 1.32 | 8.92 | 0.328 | $3.92_{\pm0.05}$ | $3.01_{\pm0.07}$ |
| AudioLCM | 3700 | 160M | 1 | 25.36 | 4.44 | 1.74 | 8.25 | 0.267 | - | - |
| AudioLCM | 3700 | 160M | 2 | 20.01 | **2.17** | 1.48 | **9.89** | 0.308 | $3.67_{\pm0.10}$ | $3.05_{\pm0.07}$ |
| AudioTurbo | $\approx 2000$ | 1.1B | 5 | 22.18 | - | 1.30 | 8.88 | - | - | - |
| AudioTurbo | $\approx 2000$ | 1.1B | 10 | 20.65 | - | 1.29 | 9.40 | - | - | - |
| AUDIODEAR $_{\text{w/o Dist.}}$ | 1700 | 191M | 1 | 22.09 | 3.82 | 1.22 | 8.07 | 0.298 | - | - |
| AUDIODEAR | 1700 | 191M | 1 | **18.67** | 2.79 | **1.06** | 9.66 | **0.334** | $4.27_{\pm0.04}$ | $3.27_{\pm0.06}$ |

using the same pre-trained CLAP model employed by IMPACT. The CLAP model used for training[4] is different from the one used for evaluation[5] to avoid taking advantage of training and evaluating with the same model. Subjective evaluation is conducted on 90 generated audio samples conditioned on the AudioCaps evaluation set prompts, using the user interface and rating criteria defined in AudioBox (Vyas et al., 2023). Each sample receives at least 9 independent ratings per subjective metric, with all annotators trained to follow the evaluation guidelines. Inference latency is measured as the number of seconds required to synthesize a batch of 10-second audio clips on a single NVIDIA Tesla V100 32GB VRAM GPU.

## 5 RESULTS AND DISCUSSIONS

We report evaluations of our proposed AUDIODEAR framework. We organize the results into system-level comparisons, ablation studies on representation distillation, analyses of alternative sampling methods, investigations of classifier-free guidance, and the impact of the number of samples used to estimate the energy-distance.

### 5.1 SYSTEM-LEVEL PERFORMANCE COMPARISONS

Table 1 shows that our one-step energy-scoring model with representation distillation achieves the strongest overall results on AudioCaps, outperforming prior fast sampling baselines such as AudioTurbo and ConsistencyTTA across FD, KL, CLAP, REL, and OVL. It approaches the quality of multi-step diffusion/flow matching models, including Tango 2 (Majumder et al., 2024), TangoFlux (Hung et al., 2024), MAGNET (Ziv et al., 2024), AudioLDM 2 (Liu et al., 2024a), Make-an-audio 2 (Huang et al., 2023), AudioMNTP (Yang et al., 2025), and IMPACT (Huang et al., 2025), falling

---

[4]https://huggingface.co/lukewys/laion_clap/blob/main/630k-audioset-fusion-best.pt
[5]https://huggingface.co/laion/clap-htsat-fused

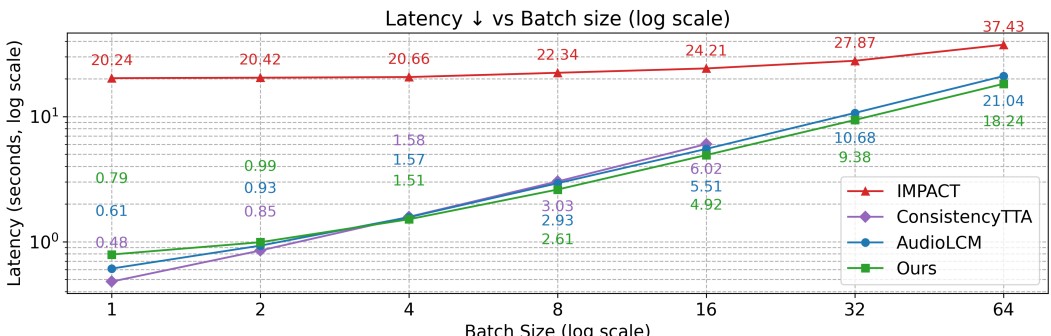

Figure 3: Inference latency of models with different batch sizes. Missing values reflect cases where the model could not accommodate the given batch size because of GPU memory constraints.

Table 2: Ablation study on distillation weights ($\lambda$).

| $\lambda$ | FD ↓ | FAD ↓ | KL ↓ | IS ↑ | CLAP ↑ |
|---|---|---|---|---|---|
| freeze | 22.79 | 4.46 | 1.24 | 7.51 | 0.288 |
| 0 | 22.09 | 3.82 | 1.22 | 8.07 | 0.298 |
| 50 | 20.52 | 2.99 | 1.12 | 8.74 | 0.316 |
| 100 | 20.04 | 3.11 | 1.12 | 8.87 | 0.321 |
| 500 | 19.62 | 2.84 | 1.10 | 8.97 | 0.322 |
| 1000 | **18.67** | **2.79** | **1.06** | **9.66** | **0.334** |
| 5000 | 19.88 | 2.98 | 1.10 | 8.76 | 0.311 |

Table 3: Ablation study on the classifier-free guidance (CFG) scale.

| CFG | FD ↓ | FAD ↓ | KL ↓ | IS ↑ | CLAP ↑ |
|---|---|---|---|---|---|
| 1.0 | 34.37 | 6.64 | 1.71 | 5.18 | 0.196 |
| 2.0 | 22.84 | 3.60 | 1.13 | 7.79 | 0.295 |
| 3.0 | 19.81 | 3.00 | 1.06 | 8.98 | 0.323 |
| 4.0 | **18.67** | **2.79** | **1.06** | **9.66** | **0.334** |
| 5.0 | 19.06 | 3.08 | 1.12 | 9.40 | 0.328 |
| 6.0 | 19.80 | 3.40 | 1.16 | 8.98 | 0.319 |

only slightly behind the two-step AudioLCM on FAD and IS, but surpassing it on both subjective evaluation metrics. Notably, our model achieves this with a single sampling step, whereas AudioLCM requires two.

Figure 3 shows the latency across different batch sizes for the state-of-the-art multi-step IMPACT model and other few-step sampling models, ConsistencyTTA, AudioLCM, and our proposed one-step sampling model AUDIODEAR, following the evaluation method of (Ziv et al., 2024). Among them, our model achieves the lowest inference latency starting from batch size 4 onward. When compared with IMPACT, the state-of-the-art 100-step diffusion-based TTA model, our approach delivers comparable objective performance, with only up to 8.6% degradation in IS, and 10.2% degradation in CLAP scores, while still achieving a roughly 25× reduction in latency for generating a 10-second audio clip.

## 5.2 REPRESENTATION DISTILLATION

The ablation study in Table 2 demonstrates the critical role of representation distillation in strengthening the one-step energy-scoring model. The setting "freeze" denotes that the transformer backbone is initialized from IMPACT and kept frozen during training, while only the lightweight energy-scoring head is optimized. Results show that freezing the IMPACT-initialized transformer backbone produces the weakest results across all metrics, confirming that fine-tuning is indispensable. Making the transformer layers trainable ($\lambda = 0$) leads to moderate improvements, and increasing the distillation weight $\lambda$ to 50 yields substantial gains, particularly in FAD, IS, and CLAP scores. Furthermore, increasing $\lambda$ further produces consistent improvements in both fidelity and semantic alignment, with the best overall results at $\lambda = 1000$, achieving the lowest FD, KL, and the highest IS and CLAP. However, setting a more aggressive $\lambda = 5000$ results in a regression in all metrics, suggesting that excessively strong distillation over-constrains the model and diminishes the benefits of distillation.

## 5.3 DIFFERENT SAMPLING METHODS

Table 4 evaluates the performance of our one-step energy-scoring method against both one-step and few-step baselines with the IMPACT-style autoregressive framework. Our proposed one-step energy-scoring method significantly outperforms other sampling baselines like Shortcut and Mean-Flow. While multi-step diffusion and flow matching models achieve strong fidelity (FD and FAD)

Table 4: Comparison of objective performance across sampling methods, including Shortcut, Mean-Flow, and our proposed AUDIODEAR model with energy-scoring, using the IMPACT-style framework. The best few-step sampling results are shown in bold.

| | # params | steps | FD ↓ | FAD ↓ | KL ↓ | IS ↑ | CLAP ↑ |
|---|---|---|---|---|---|---|---|
| Diffusion | 193M | 100 | 15.25 | 1.26 | 1.06 | 10.57 | 0.372 |
| | 193M | 4 | 138.95 | 34.34 | 4.90 | 1.52 | -0.049 |
| | 193M | 1 | 128.47 | 34.44 | 4.94 | 1.18 | -0.047 |
| Flow matching | 193M | 100 | 15.65 | 1.78 | 1.05 | 10.33 | 0.377 |
| | 193M | 4 | 69.26 | 14.91 | 2.16 | 3.60 | 0.179 |
| | 193M | 1 | 126.44 | 43.79 | 4.17 | 1.02 | -0.057 |
| MeanFlow | 194M | 4 | 34.33 | 11.19 | 1.51 | 5.78 | 0.252 |
| | 194M | 1 | 79.46 | 13.52 | 3.81 | 2.34 | 0.080 |
| Shortcut Model | 194M | 4 | 63.99 | 12.39 | 2.32 | 3.55 | 0.172 |
| | 194M | 1 | 98.12 | 27.33 | 4.12 | 1.27 | -0.073 |
| Energy-scoring (Ours) | 191M | 1 | 22.09 | 3.82 | 1.22 | 8.07 | 0.298 |
| Energy-scoring + distill (Ours) | 191M | 1 | **18.67** | **2.79** | **1.06** | **9.66** | **0.334** |

and high semantic alignment (CLAP), their quality degrades sharply when reduced to one or a few steps. In contrast, our energy-scoring approach maintains substantially lower FD and FAD scores and higher IS and CLAP values in the one-step setting, indicating better perceptual quality and semantic relevance. The distillation-enhanced variant achieves the best one-step results overall, with objective scores relatively comparable to the 100-step diffusion baseline, demonstrating that representation-level guidance from the diffusion-trained teacher effectively narrows the quality gap while retaining the efficiency of one-step sampling.

## 5.4 CLASSIFIER-FREE GUIDANCE

Table 3 examines the effect of varying the classifier-free guidance (CFG) scale on our energy-scoring model with representation distillation. The results show a clear trend where increasing CFG from $1.0$ to $4.0$ progressively improves performance across objective metrics, with the best overall performance achieved at CFG $= 4.0$. Lower CFG values, such as $1.0$, result in substantially degraded semantic alignment and audio quality, while excessively high values beyond $4.0$ lead to slight degradation, suggesting an optimal balance between guidance strength and audio quality at CFG $= 4.0$.

Table 5: Ablation study on the number of samples used to calculate the energy distance for training.

| num samples $m$ | FD ↓ | FAD ↓ | KL ↓ | IS ↑ | CLAP ↑ |
|---|---|---|---|---|---|
| $m = 2$ | 18.67 | 2.79 | **1.06** | **9.66** | **0.334** |
| $m = 3$ | 18.32 | 2.68 | 1.11 | 9.24 | 0.322 |
| $m = 4$ | **18.13** | **2.53** | 1.09 | 9.19 | 0.322 |

## 5.5 NUMBER OF SAMPLES FOR ENERGY-DISTANCE ESTIMATION

During training, two random samples produced by the model $x_1$ and $x_2$ are used to calculate the training objective as shown in Equation 3. More generally, a larger number of samples can be drawn to estimate the energy-distance via the extended form of Equation 9 in Appendix B. Table 5 investigates the effect of varying the number of samples $m$ used during training. Increasing $m$ from 2 to 4 progressively reduces both FD and FAD scores, suggesting improved fidelity. Specifically, FD decreases from $18.67$ at $m = 2$ to $18.13$ at $m = 4$, while FAD drops from $2.79$ to $2.53$. However, this gain comes with nuanced trade-offs: although FD and FAD improve, the KL divergence slightly worsens when moving from $m = 2$ to higher sample counts, and the IS peaks at $m = 2$ with $9.66$ before dropping modestly at larger values of $m$. Similarly, CLAP scores are highest with $m = 2$ but decrease at both $m = 3$ and $m = 4$. Overall, these findings suggest that while larger sample sizes enhance fidelity, the setting of $m = 2$ provides the best balance, yielding the strongest semantic alignment and generative diversity.

Table 6: Comparing AR steps ($r$), sampling steps ($n$), and objective performance between IMPACT and our AUDIODEAR model. "FLOPs" stands for the number of floating-point operations.

| Model | $r$ | $n$ | FD↓ | FAD↓ | KL↓ | IS↑ | CLAP↑ | Lat. (s) | FLOPs |
|---|---|---|---|---|---|---|---|---|---|
| (a) IMPACT | 64 | 100 | 15.25 | 1.26 | 1.06 | 10.57 | 0.372 | 20.24 | 1.11e13 |
| (b) IMPACT | 4 | 100 | 19.93 | 3.49 | 1.15 | 8.65 | 0.325 | 1.21 | 2.64e12 |
| (c) IMPACT | 4 | 75 | 21.60 | 3.63 | 1.27 | 8.37 | 0.313 | 1.10 | 2.12e12 |
| (d) IMPACT | 4 | 50 | 36.30 | 7.63 | 1.73 | 6.51 | 0.230 | 0.62 | 1.60e12 |
| (e) IMPACT | 64 | 1 | 128.47 | 34.44 | 4.94 | 1.18 | -0.047 | 0.83 | 9.00e12 |
| (f) IMPACT | 6 | 50 | 24.54 | 3.83 | 1.47 | 7.70 | 0.273 | 1.20 | 1.88e12 |
| AudioDEAR (ours) | 64 | 1 | 18.67 | 2.79 | 1.06 | 9.66 | 0.334 | 0.79 | 8.99e12 |

# 6    LATENCY-QUALITY TRADEOFF OF IMPACT

To assess the effect of AR decoding steps and sampling steps on IMPACT, we compare 6 configurations, models (a) to (f), in Table 6. Across all settings, reducing either $r$ or $n$ consistently degrades the objective metrics. With $r = 64$, comparing IMPACT models (a) and (e) highlights this tradeoff clearly: model (e) achieves very low latency by using only one sampling step ($n = 1$), but the overall objective performance collapses. Notably, AudioDEAR also uses only one sampling step yet maintains competitive objective metrics, underscoring IMPACT's limitations under this one-step sampling configuration. When $r = 4$, IMPACT models (b), (c), and (d) show progressively worse objective performance as the sampling steps ($r$) decrease. Although IMPACT model (d) achieves latency comparable to our AUDIODEAR model, its performance on objective metrics remains suboptimal. Overall, simply adjusting the number of AR decoding steps or sampling steps is insufficient for IMPACT to approach the performance of our AudioDEAR model.

# 7    CONCLUSIONS AND FUTURE WORK

We introduce a one-step TTA framework trained with an energy-distance objective and representation distillation from a diffusion-trained teacher. By eliminating the need for multiple sampling steps at each decoding iteration, our method achieves 25× faster inference than the state-of-the-art TTA model, IMPACT, while maintaining strong audio fidelity and semantic relevance. Our extensive experiments on AudioCaps show significant gains over existing strong few-step sampling baselines and a narrowed gap to multi-step diffusion systems. These results demonstrate that combining energy-distance training with representation-level guidance offers an effective recipe for low-latency, high-quality audio generation. In future work, we aim to further reduce AR steps to push the limits of low-latency audio generation.

ETHICS STATEMENT

This research focuses on developing a one-step TTA framework for efficient, high-quality audio generation, with potential applications in creative and beneficial domains such as gaming, advertising, and virtual reality. Our model has not been trained or optimized for reproducing identifiable voices, nor for generating harmful or discriminatory content. The subjective evaluation was conducted exclusively by independent, full-time domain experts within the organization. These experts had no conflicts of interest, participated voluntarily, and applied established ethical research standards to ensure objectivity and reliability in the assessment.

REPRODUCIBILITY STATEMENT

The methodology for our proposed one-step TTA generation framework is detailed in Section 3 of the main paper, including the masked autoregressive energy-scoring framework and the iterative parallel decoding inference process. The training and evaluation details are provided in Section 4, which includes information on the datasets used, model configurations, and the metrics for both objective and subjective assessment. The specific hyperparameters used, such as the learning rate, batch size, masking rate, and distillation weight $\lambda$, are also listed in the Section 4.2. The appendices complement the main text with further technical details. Appendix A presents the complete proof of the energy-distance objective. Appendix E describes the architecture of the energy-scoring module. Appendix H offers a comprehensive list of the training data combinations used for each model discussed in the paper. Appendix I provides structural diagrams for the training and inference pipeline.

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

## A  ENERGY-DISTANCE

The following content lists out the definitions and theorems required to prove Corollary 1, stated as follows.

**Corollary 1.** *Let $X$ and $Y$ be independent random vectors in $\mathbb{R}^d$ with distributions $P$ and $Q$, respectively. Then*

$$2\mathbb{E}[\|X - Y\|] - \mathbb{E}[\|X - X'\|] - \mathbb{E}[\|Y - Y'\|] \geq 0,$$

*where $X'$ and $Y'$ are independent copies of $X$ and $Y$, respectively. Equality holds if and only if $P = Q$.*

The proof begins by recalling the notion of a negative definite kernel.

**Definition 1** (Negative definite kernel). *Let $\mathcal{X}$ be a nonempty set. A symmetric function*

$$g : \mathcal{X} \times \mathcal{X} \to \mathbb{R}$$

*is called* negative definite *if for every $n \in \mathbb{N}$, every choice of points $x_1, \ldots, x_n \in \mathcal{X}$, and every set of real coefficients $r_1, \ldots, r_n$ satisfying*

$$\sum_{j=1}^{n} r_j = 0,$$

*the inequality*

$$\sum_{j=1}^{n} \sum_{k=1}^{n} r_j r_k \, g(x_j, x_k) \leq 0$$

*holds.*

**Definition 2** (Strictly negative definite kernel). *A kernel $g$ is said to be* strictly negative definite *if it is negative definite and the inequality above is strict whenever the coefficients $(r_1, \ldots, r_n)$ are not identically zero.*

**Proposition 1.** *The Euclidean distance*

$$g(x, y) = \|x - y\|, \qquad x, y \in \mathbb{R}^d,$$

*is a strictly negative definite kernel. This is proved in the Appendix of (Székely & Rizzo, 2005).*

**Interpretation.** Proposition 1 asserts that for any finite collection of points $x_1, \ldots, x_n \in \mathbb{R}^d$ and any coefficients $r_1, \ldots, r_n \in \mathbb{R}$ with $\sum_{j=1}^{n} r_j = 0$, one has

$$\sum_{j=1}^{n} \sum_{k=1}^{n} r_j r_k \, \|x_j - x_k\| < 0,$$

unless $r_1 = \cdots = r_n = 0$. By definition, it is not hard to derive that

$$\sum_{j=1}^{n} \sum_{k=1}^{n} r_j r_k \, \|x_j - x_k\| \; \leq \; 0,$$

whenever $\sum_{j=1}^{n} r_j = 0$, where the equality holds if and only if $r(x) = 0$. This property establishes that the Euclidean distance, when viewed as a kernel, induces quadratic forms that are nonpositive under zero-sum weighting and strictly negative unless the weighting is trivial. This structural property is the key ingredient in the derivation of the energy-distance between probability measures, which underlies Corollary 1.

**Theorem 1.** *For any two independent random variables $X \sim P$ and $Y \sim Q$, we have*

$$2\mathbb{E}\left[g(X,Y)\right] - \mathbb{E}\left[g(X,X')\right] - \mathbb{E}\left[g(Y,Y')\right] \; \geq \; 0,$$

*where $g$ is the Euclidean distance, $X'$ and $Y'$ are independent copies of $X$ and $Y$, respectively. The equality holds if and only if $P = Q$.*

The proof of Theorem 1 builds upon the proof of Theorem 1 in (Székely & Rizzo, 2005), presented here in an expanded and more detailed form.

*Proof.* Assume the expectations in the statement are finite (this is ensured, e.g., by $\mathbb{E}\|X\| + \mathbb{E}\|Y\| < \infty$ when $g(x,y) = \|x - y\|$). Let $\mu$ and $\nu$ denote the laws of $X$ and $Y$, respectively, and fix a probability measure $W$ dominating both $\mu$ and $\nu$. Define

$$r(x) \;=\; \frac{d\mu}{dW}(x) - \frac{d\nu}{dW}(x), \qquad \text{so that} \qquad \int_{\mathcal{X}} r(x)\, dW(x) = 0.$$

By independence, the joint law of a pair is the product measure of their marginals. Combined with Fubini-Tonelli theorem, the three expectations can be written as:

$$\mathbb{E}\left[g(X,X')\right] = \int_{\mathcal{X}} \int_{\mathcal{X}} g(x,y)\, d\mu(x)\, d\mu(y), \quad \mathbb{E}\left[g(Y,Y')\right] = \int_{\mathcal{X}} \int_{\mathcal{X}} g(x,y)\, d\nu(x)\, d\nu(y),$$

$$\mathbb{E}\left[g(X,Y)\right] = \int_{\mathcal{X}} \int_{\mathcal{X}} g(x,y)\, d\mu(x)\, d\nu(y).$$

Since $d\mu = \frac{d\mu}{dW}\, dW$ and $d\nu = \frac{d\nu}{dW}\, dW$, we can express these as

$$E[g(X,X')] = \int_{\mathcal{X}} \int_{\mathcal{X}} g(x,y)\, \frac{d\mu}{dW}(x)\, \frac{d\mu}{dW}(y)\, dW(x)\, dW(y),$$

$$E[g(Y,Y')] = \int_{\mathcal{X}} \int_{\mathcal{X}} g(x,y)\, \frac{d\nu}{dW}(x)\, \frac{d\nu}{dW}(y)\, dW(x)\, dW(y),$$

$$E[g(X,Y)] = \int_{\mathcal{X}} \int_{\mathcal{X}} g(x,y)\, \frac{d\mu}{dW}(x)\, \frac{d\nu}{dW}(y)\, dW(x)\, dW(y).$$

Therefore,

$$2E[g(X,Y)] - E[g(X,X')] - E[g(Y,Y')]$$
$$= \int_{\mathcal{X}} \int_{\mathcal{X}} g(x,y)\left(2\frac{d\mu}{dW}(x)\frac{d\nu}{dW}(y) - \frac{d\mu}{dW}(x)\frac{d\mu}{dW}(y) - \frac{d\nu}{dW}(x)\frac{d\nu}{dW}(y)\right) dW(x)\, dW(y).$$

Since $g(x,y)$ is symmetric, we may replace the middle term in parentheses by

$$-\left(\frac{d\mu}{dW}(x) - \frac{d\nu}{dW}(x)\right)\left(\frac{d\mu}{dW}(y) - \frac{d\nu}{dW}(y)\right).$$

Thus,

$$2E[g(X,Y)] - E[g(X,X')] - E[g(Y,Y')] = -\int_{\mathcal{X}} \int_{\mathcal{X}} g(x,y)\, r(x)\, r(y)\, dW(x)\, dW(y).$$

Now set $g(x,y) = \|x - y\|$. By Proposition 1, $g$ is a strictly negative definite kernel on $\mathbb{R}^d$. Therefore, for any $r$ with $\int_{\mathcal{X}} r(x)\, dW(x) = 0$,

$$\int_{\mathcal{X}} \int_{\mathcal{X}} g(x,y)\, r(x)\, r(y)\, dW(x)\, dW(y) \; \leq 0,$$

with equality if and only if $r(x) = 0$ $W$-a.s. Consequently,

$$2\mathbb{E}\left[g(X,Y)\right] - \mathbb{E}\left[g(X,X')\right] - \mathbb{E}\left[g(Y,Y')\right] \; \geq 0,$$

with equality if and only if $r(x) = 0$ $W$-a.s., i.e., $\mu = \nu$ and hence $P = Q$. This proves the theorem. $\qquad\square$

## B  ENERGY-DISTANCE LOSS CALCULATION

In this section, we rewrite the energy-distance in Equation 1 in the form of an estimation with a finite number of samples as shown in Equation 7,

$$\mathcal{E}(P,Q) = \frac{2}{mn} \sum_{i=1}^{m} \sum_{j=1}^{n} \|X_i - Y_j\| - \frac{1}{m(m-1)} \sum_{\substack{i,j=1 \\ i \neq j}}^{m} \|X_i - X_j\| - \frac{1}{n(n-1)} \sum_{\substack{i,j=1 \\ i \neq j}}^{n} \|Y_i - Y_j\| \quad (7)$$

where $m$ is the number of samples drawn from distribution $P$, and $n$ is the number of samples drawn from distribution $Q$. In the context of model training, the term $\|Y_i - Y_j\|$ is a constant and can be ignored during optimization. Thus, Equation 7 can be rewritten into:

$$\widetilde{\mathcal{E}}(P,Q) = \frac{2}{mn} \sum_{i=1}^{m} \sum_{j=1}^{n} \|X_i - Y_j\| - \frac{1}{m(m-1)} \sum_{\substack{i,j=1 \\ i \neq j}}^{m} \|X_i - X_j\|. \quad (8)$$

More specifically, for each data point $y$ drawn from distribution $Q$, the energy-distance can be estimated by drawing $m$ samples $x_1, x_2, \cdots, x_m \sim P_\theta$ and calculating the following equation:

$$\mathcal{L}_{\text{energy}} = \frac{2}{m} \sum_{i=1}^{m} \|x_i - y\| - \frac{1}{m(m-1)} \sum_{\substack{i,j=1 \\ i \neq j}}^{m} \|x_i - x_j\|, \quad (9)$$

where $m = 2$ reduces to Equation 3.

## C  TEXT EMBEDDINGS

Table 7 examines how different text embedding choices affect the performance of our one-step energy-scoring model with representation distillation. The best overall results are achieved when using a combination of CLAP and Flan-T5 embeddings. The model's performance remains strong even when the CLAP embeddings are removed, with only a negligible drop in metrics. This suggests that the framework's training does not significantly benefit from using CLAP embeddings to improve its CLAP metric score. It is important to note that the CLAP model used for training and inference is different. Conversely, the most significant performance drop occurs when only CLAP embeddings are used. In this scenario, the FD and FAD metrics substantially worsen, and KL, IS, and CLAP also degrade.

Table 7: Ablation study on the choice of text embeddings.

| text embeddings | FD ↓ | FAD ↓ | KL ↓ | IS ↑ | CLAP ↑ |
|---|---|---|---|---|---|
| CLAP + Flan-T5 | **18.67** | **2.79** | **1.06** | **9.66** | **0.334** |
| only Flan-T5 | 18.79 | 2.76 | 1.08 | 9.57 | 0.331 |
| only CLAP | 20.10 | 3.21 | 1.22 | 9.01 | 0.307 |

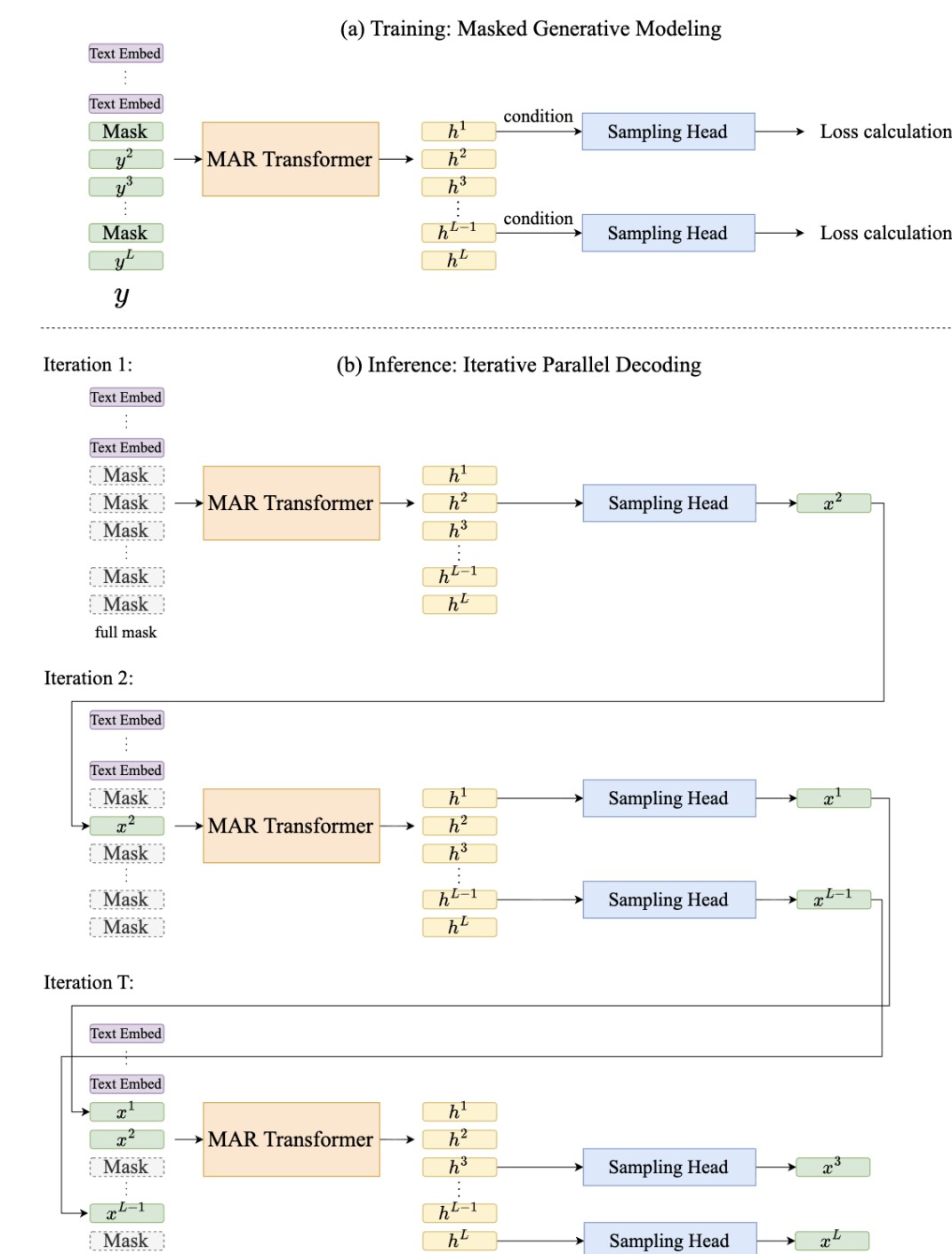

Figure 4: Illustration of a mask autoregressive continuous sampling framework. (a) Training pipeline with masked generative modeling. (b) Inference pipeline with iterative parallel decoding.

# D  MASKED AUTOREGRESSIVE CONTINUOUS SAMPLING

Figure 4 illustrates the masked autoregressive continuous sampling framework mentioned in Section 3.1. As shown in Figure 4(a), training is carried out by masked generative modeling, which randomly masks a portion of VAE latents $y$, and makes the framework predict the masked positions, with the loss being the loss of the corresponding sampling method, such as diffusion, flow matching, or energy-scoring. As shown in Figure 4(b), inference is performed by iterative parallel decoding.

Starting with a full sequence of mask tokens in the first iteration, a random set of positions is selected to be generated. The generated latents will serve as input during the next iteration. This process repeats until all positions are generated.

## E  ENERGY-SCORING MODULE

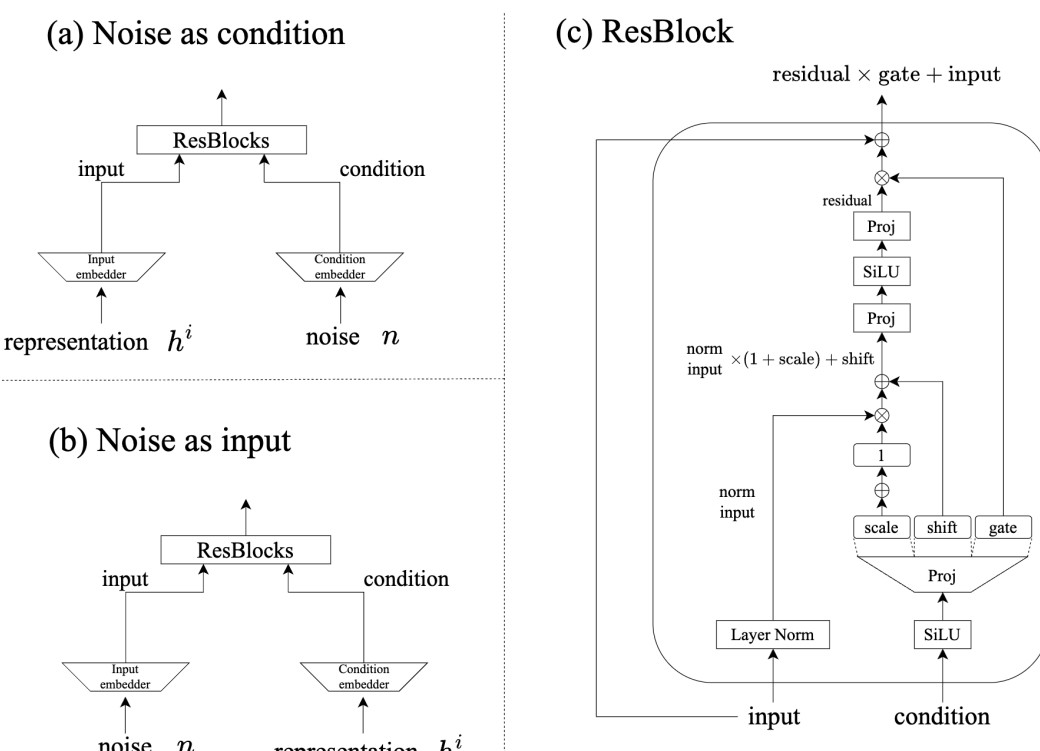

Figure 5: Configurations for the energy-scoring module. (a) Noise as condition. Contextual representation as input. (b) Noise as input. Contextual representation as condition. (c) ResBlock architecture.

Figure 5 depicts the design alternatives for the energy-scoring module, highlighting two different configurations for incorporating noise. In Figure 5 (a), the contextual representation $h^i$ is used as the main input to the ResBlocks, while the sampled noise vector $n$ is treated as the conditioning signal, passed into the ResBlocks and incorporated with adaptive layer normalization (ada-LN). In Figure 5 (b), the sampled noise vector $n$ is used as the main input to the ResBlocks, while the contextual representation $h^i$ is treated as the conditioning signal, passed into the ResBlocks and incorporated with adaptive layer normalization (ada-LN).

Table 8: Ablation study on the configuration of the energy-scoring module.

| configuration | FD $\downarrow$ | FAD $\downarrow$ | KL $\downarrow$ | IS $\uparrow$ | CLAP $\uparrow$ |
|---|---|---|---|---|---|
| (a) Noise as condition | 28.32 | 4.95 | 1.31 | 7.19 | 0.265 |
| (b) Noise as input | **22.09** | **3.82** | **1.22** | **8.07** | **0.298** |

Table 8 ablates the different designs for the configuration of the energy-scoring module (no distillation techniques are applied). Using noise as the primary input (configuration (b)) consistently outperforms the alternative of treating noise as a conditioning signal (configuration (a)) across all evaluation metrics. Specifically, configuration (b) achieves substantially lower FD (22.09 vs. 28.32) and FAD (3.82 vs. 4.95), alongside improvements in KL divergence and CLAP similarity, indicating both better fidelity and stronger semantic alignment. These results confirm that structuring the module with noise as the main input while leveraging contextual representations as the conditioning

pathway yields a more effective mapping from noise to audio latents, thereby improving one-step TTA generation quality.

# F SUBJECTIVE EVALUATION

In this section, we present the results of a subjective evaluation of text-relevance (REL) and overall audio quality (OVL) on 90 AudioCaps samples from the evaluation set. We compare our proposed AUDIODEAR framework with prior few-step sampling baselines and the state-of-the-art IMPACT system. Table 9 reports mean ratings along with their standard deviations, standard errors, and 95% confidence intervals.

Table 9: Performance and statistical values for the text-relevance (REL) and overall audio quality (OVL) metrics on 90 audio samples with text prompts sampled from the AudioCaps evaluation set. "stdev" stands for standard deviation. "stderr" stands for standard error. "CI" stands for confidence intervals.

| Method | REL | | | | OVL | | | |
|---|---|---|---|---|---|---|---|---|
| | mean | stdev | stderr | CI | mean | stdev | stderr | CI |
| Ground Truth | 4.45 | 0.27 | 0.09 | [4.28, 4.62] | 3.68 | 0.24 | 0.08 | [3.53, 3.83] |
| Tango 2 | 4.07 | 0.26 | 0.08 | [3.91, 4.23] | 3.42 | 0.28 | 0.09 | [3.25, 3.59] |
| EzAudio-XL | 4.03 | 0.25 | 0.08 | [3.88, 4.18] | 3.31 | 0.23 | 0.07 | [3.17, 3.45] |
| IMPACT | 4.38 | 0.31 | 0.10 | [4.19, 4.57] | 3.47 | 0.29 | 0.09 | [3.29, 3.65] |
| ConsistencyTTA | 3.92 | 0.17 | 0.05 | [3.81, 4.03] | 3.01 | 0.21 | 0.07 | [2.88, 3.14] |
| AudioLCM | 3.67 | 0.33 | 0.10 | [3.47, 3.87] | 3.05 | 0.21 | 0.07 | [2.92, 3.18] |
| AUDIODEAR | 4.27 | 0.14 | 0.04 | [4.18, 4.36] | 3.27 | 0.19 | 0.06 | [3.15, 3.39] |

Among the existing few-step sampling models, AUDIODEAR attains a REL score of 4.27, nearly closing the gap to IMPACT while clearly outperforming other baselines. In particular, AU-DIODEAR surpasses ConsistencyTTA (3.92) and AudioLCM (3.67) in text-relevance by large margins, with confidence intervals that do not overlap. This indicates that incorporating an energy-scoring objective with representation-level distillation substantially yields good semantic consistency with the conditioning text.

For perceived audio quality, IMPACT again leads with a mean OVL score of 3.47. AUDIODEAR achieves 3.27, outperforming both ConsistencyTTA (3.01) and AudioLCM (3.05). While a modest gap remains relative to IMPACT, the statistical bounds confirm that AUDIODEAR yields consistently higher perceptual quality than other few-step sampling methods, validating the effectiveness of our one-step synthesis design. Importantly, this gain is achieved while retaining a one-step sampling budget, offering a significantly faster alternative to multi-step autoregressive diffusion models.

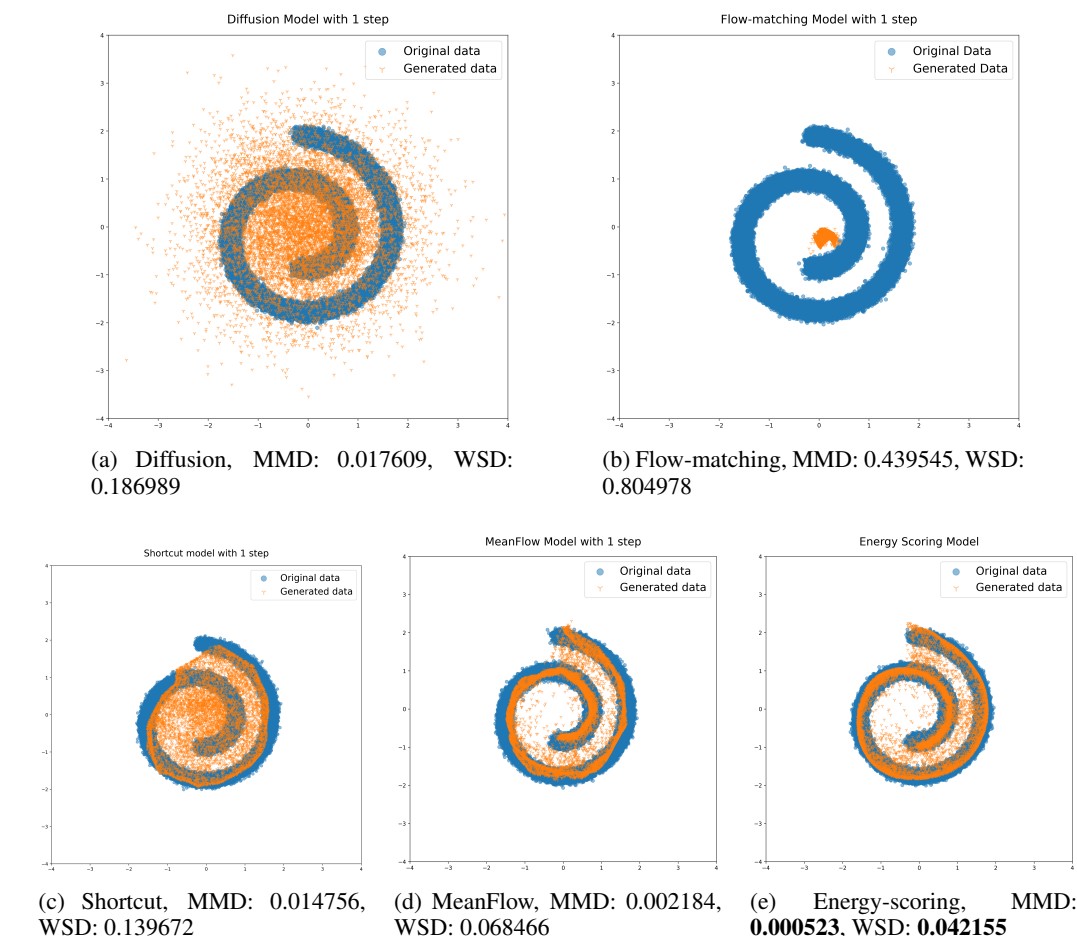

Figure 6: Comparisons of different continuous sampling methods with a toy example of a Swiss roll. Maximum mean discrepancy (MMD, ↓) and Wasserstein distance (WSD, ↓) are used to measure the distribution-wise difference between the original data and the generated data for each model.

## G   TOY EXAMPLE FOR DIFFERENT CONTINUOUS SAMPLING METHODS

To elucidate the distinctions between alternative one-step continuous sampling approaches, we present a toy experiment. Figure 6 reports both qualitative and quantitative comparisons across different methods: Diffusion (Ho et al., 2020), Flow matching (Lipman et al., 2023), Shortcut (Frans et al., 2025), MeanFlow (Geng et al., 2025), and our proposed Energy-scoring method. We adopt the Swiss roll dataset, where the ground-truth original data distribution is shown in blue and the generated samples are shown in orange. This visualization highlights how closely each method recovers the underlying geometry of the data manifold. Beyond qualitative inspection, we quantitatively assess distributional fidelity using two widely recognized metrics: maximum mean discrepancy (MMD) and Wasserstein distance (WSD). In both cases, lower values indicate a tighter alignment between the synthetic (orange) and real (blue) distributions.

The one-step diffusion method results in the generated distribution resembling the source Gaussian distribution. The one-step flow matching method results in the mean point of the target distribution, because the starting points of the ODE tend to have the directions of the velocity pointing to the mean of the target distribution. The one-step Shortcut method results in a distribution with a contour similar to the target distribution, but fails to model the target distribution accurately. The one-step MeanFlow method and our Energy-scoring method both generate data with a shape similar to the original data distribution, having sharper alignment with the spiral geometry. However, comparing Figure 6(d) and Figure 6(e), it is shown that MeanFlow fails to sufficiently cover the full spread of the

original data, while our Energy-scoring method has broader coverage of the spiral data distribution. The MMD and WSD metrics also verify that our Energy-scoring method aligns better with the original data.

# H   DATASET INFORMATION

Table 10: Training data of each text-to-audio generation model. Any dataset that is involved during any training phase, including pre-training and fine-tuning, will be listed out in this table, regardless of whether the full set of the dataset is used.

| Models | Data Configuration |
|---|---|
| Tango-full-ft | AS+AC+FS+BBC+US+MI+MC+GMG+ESC50 |
| Tango-AF&AC-FT-AC | AFAS+AC |
| Tango 2 | AS+AC+FS+BBC+US+MI+MC+GMG+ESC50+AA |
| TangoFlux | AC+WC |
| EzAudio-L (24kHz) | AS+AACD+ASQC+ASSLGC+AC |
| EzAudio-XL (24kHz) | AS+AACD+ASQC+ASSLGC+AC |
| MAGNET-L | AS+BBC+AC+Cv2+VGG+FSD50K+FTUS+SGE+WSE+PM |
| Make-an-Audio 2 | AS+AC+WC+AASE+ASTK+ESC50+FSD50K+MACS+ES+US+WT+TUT |
| AudioLDM2-full | AS+AC+WC+VGG+FMA+MSD+LJS+GGS |
| AudioMNTP | AC+WC |
| IMPACT | AC+WC |
| ConsistencyTTA | AC |
| AudioLCM | AS+AC+WC+AASE+ASTK+ESC50+FSD50K+MACS+ES+US+WT+TUT |
| AudioTurbo | AC+MACS+Cv2+ESC50+US+MI+GMG+WC |
| AUDIODEAR | AC+WC+AS |

**Dataset Abbreviations:**

- **AA:** Audio-alpaca [6]
- **AACD:** Auto-ACD (Sun et al., 2024a)
- **AASE:** Adobe Audition Sound Effects [7]
- **AC:** AudioCaps Kim et al. (2019)
- **AFAS:** AF-AudioSet
- **AS:** AudioSet (Gemmeke et al., 2017)
- **ASQC:** AS-Qwen-Caps
- **ASSLGC:** AS-SL-GPT4-Caps
- **ASTK:** Audiostock [8]
- **BBC:** BBC sound effects
- **Cv2:** Clotho v2 (Drossos et al., 2020)
- **ES:** Epidemic Sound [9]
- **ESC50:** Environmental Sound Classification (Piczak, 2015)
- **FMA:** Free Music Archive (Defferrard et al., 2016)
- **FS:** Freesound Dataset [10]
- **FSD50K:** Freesound Dataset 50k citepfonseca2021fsd50k [11]

---

[6] https://huggingface.co/datasets/declare-lab/audio-alpaca
[7] https://www.adobe.com/products/audition/offers/adobeauditiondlcsfx.html
[8] https://audiostock.net/
[9] https://www.epidemicsound.com/
[10] https://freesound.org/
[11] https://zenodo.org/records/4060432

- **FTUS:** Free To Use Sounds
- **GGS:** GigaSpeech (Chen et al., 2021)
- **GMG:** Gtzan Music Genre
- **LJS:** LJSpeech [12]
- **MACS:** MACS (Martín-Morató & Mesaros, 2021)
- **MC:** MusicCaps
- **MI:** Musical Instrument
- **MSD:** Million Song Dataset (Bertin-Mahieux et al., 2011)
- **PM:** Paramount Motion
- **SGE:** Sonniss Game Effects
- **TUT:** TUT acoustic scene Mesaros et al. (2016)
- **US:** Urban Sound (Salamon et al., 2014)
- **VGG:** VGG-Sound
- **WC:** WavCaps (Mei et al., 2024)
- **WSE:** WeSoundEffects
- **WT:** WavText5Ks (Deshmukh et al., 2022)

---

[12]https://keithito.com/LJ-Speech-Dataset/

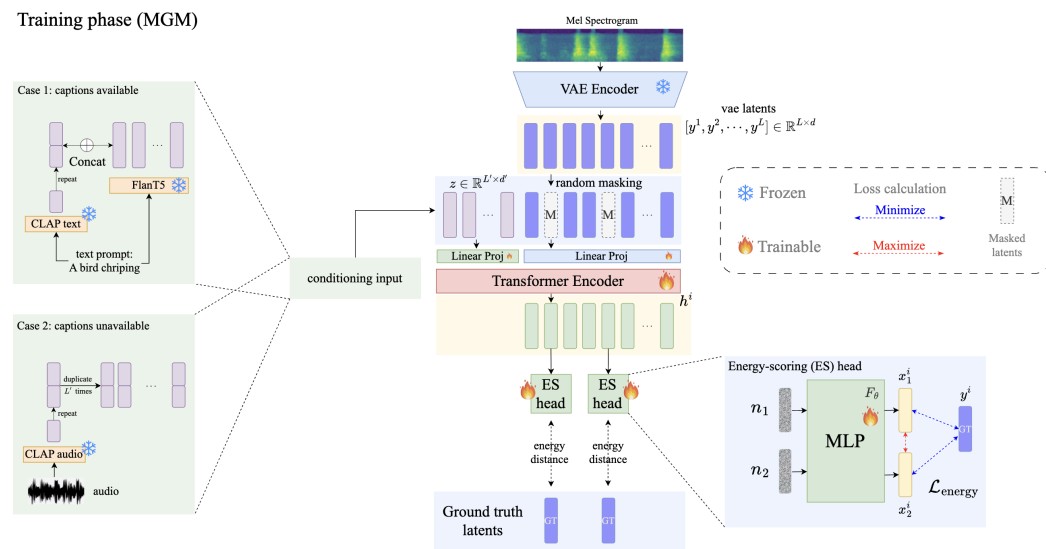

Figure 7: Illustration of the training framework with masked generative modeling by energy-scoring.

# I OVERALL STRUCTURE

As shown in Figure 7, during training, the transformer receives two types of inputs: conditioning embeddings and VAE latents. The VAE latents are produced by encoding Mel spectrograms with a pre-trained VAE encoder from (Liu et al., 2023). The conditioning embeddings are constructed differently depending on the caption availability for each audio sample. To clearly describe this process, we distinguish between two cases in Section I.1.

## I.1 CONDITIONING EMBEDDINGS

**Case 1: Captioned audio (AudioCaps and WavCaps).** When captions are available, we use both the CLAP text encoder and the Flan-T5 encoder. The CLAP text encoder outputs a single 512-dimensional embedding, whereas the Flan-T5 encoder outputs 77 embeddings of dimension 1024. To align these representations, we repeat the CLAP text embedding once along its embedding dimension, producing a 1024-dimensional vector. Concatenating this repeated CLAP embedding with the Flan-T5 embeddings yields a conditioning sequence of length 78.

**Case 2: Uncaptioned audio (AudioSet).** When captions are unavailable, we still maintain a conditioning sequence of length 78. In this case, a single 512-dimensional CLAP audio embedding is extracted for each audio clip and expanded to 1024 dimensions by repeating it once along the sequence length dimension. This 1024-dimensional vector is duplicated 78 times to form the conditioning sequence.

## I.2 TRAINING (MASKED GENERATIVE MODELING)

As shown in Figure 7, during masked generative modeling, a subset of the VAE latents is randomly masked. Both the masked latent sequence and the conditioning embeddings are passed through linear projection layers to match the transformer's hidden dimension. For each masked position, the energy-scoring head takes the corresponding transformer output as input and uses two sampled noise vectors to compute the energy distance objective described in Eq. (3). Most importantly, all models reported in the paper are trained on a unified mixture of AudioCaps, WavCaps, and AudioSet. No model is trained on individual datasets, and no separate system configurations based on different dataset combinations are used.

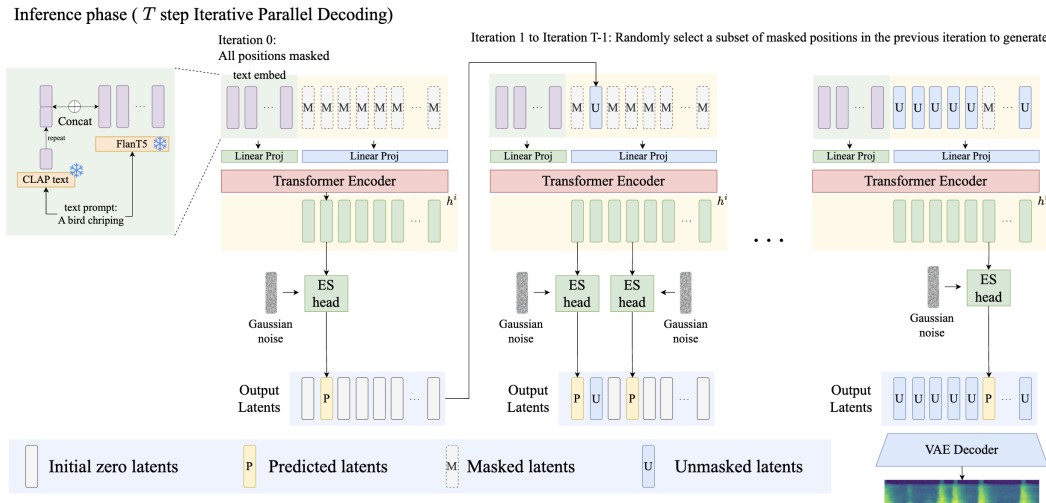

Figure 8: Illustration of the inference phase with iterative parallel decoding with an energy-scoring framework. "ES head" denotes the energy-scoring head.

### I.3 INFERENCE (ITERATIVE PARALLEL DECODING)

As shown in Figure 8, during inference, we use iterative parallel decoding to gradually construct the full latent sequence as used in (Huang et al., 2025). In the first decoding iteration, the model receives the text embeddings together with a fully masked latent sequence. In each iteration, the energy-scoring head predicts a randomly selected subset of latent positions. These predicted latents are inserted back into their corresponding positions in the input sequence, replacing the masked tokens and serving as the unmasked inputs for the next iteration. Throughout the decoding process, all positions are eventually generated. Once the full latent sequence is completely generated, the VAE decoder converts it back into a Mel spectrogram to produce the output audio.

## J    LLM USAGE

Large Language Models (LLMs) were used only for minor editing and polishing of the writing. They were not involved in generating ideas, conducting experiments, creating figures, or contributing substantive content.

