# OpenReview forum: "Fast Text-to-Audio Generation with One-Step Sampling via Energy-Scoring and Auxiliary Contextual Representation Distillation"
_ICLR.cc/2026/Conference — Submitted to ICLR 2026_

### Official Review · Reviewer_Rb5X · 2025-10-22

**Soundness:** 3
**Presentation:** 1
**Contribution:** 2
**Rating:** 4
**Confidence:** 4

**Summary:**

The paper proposes a one-step text-to-audio generation framework that builds upon the IMPACT model. The main contribution lies in replacing the diffusion-based loss with an energy-distance objective and introducing a representation-level distillation process to enhance one-step generation quality. While the idea of simplifying generation through distillation is relevant and aligns with current trends in efficient diffusion or flow-matching models, the overall contribution appears incremental. Moreover, the paper’s presentation lacks clarity, making it difficult to follow the relationship between the proposed loss, model inputs and outputs, and overall architecture.

**Strengths:**

•  The work explores a distillation-based strategy to enhance the performance of a one-step generation framework, which could reduce computational cost and inference time.

•  The paper includes a wide range of experiments examining the influence of different distillation weights and classifier-free guidance strengths.

•  The idea of leveraging an energy-distance objective as an alternative to diffusion loss is conceptually interesting and potentially valuable for fast generation research.

**Weaknesses:**

•  The writing and structure of the paper need substantial improvement. The current format makes it difficult to follow the key ideas; for instance, the introduction includes too many technical details that belong to the related work or methodology sections.

•  The method section is overly complicated and lacks a clear explanation of the model architecture—no structural figure is provided, and the correspondence between equations and model components is unclear.

•  The dataset processing description is confusing. In particular, the approach to handling audio longer than 10 seconds and the captioning process for AudioSet (which contains only label data) are not well explained.

•  The subjective quality of the generated audio in the demos does not align with the quantitative results reported in the tables, and the improvements over state-of-the-art consistency-based models are not convincing.

•  The ablation study on sampling methods seems unconvincing, as existing flow-matching or consistency models should achieve comparable results with minimal sampling steps.

**Questions:**

•  How are captions handled when audio clips longer than 10 seconds are randomly cropped into 10-second segments?

•  How were captions generated for the 500-hour AudioSet subset, given that AudioSet only provides label-level annotations?

•  How many participants were involved in the subjective evaluation on the 90 generated samples?

•  Which specific flow-matching approach or implementation was used for the experiments reported in Table 4?

---

> ### Author Response · Authors · 2025-11-25
>
> **Q: The writing and structure of the paper need substantial improvement. The current format makes it difficult to follow the key ideas; for instance, the introduction includes too many technical details that belong to the related work or methodology sections.
> The method section is overly complicated and lacks a clear explanation of the model architecture—no structural figure is provided, and the correspondence between equations and model components is unclear.**
>
> Ans: We thank the reviewer for this feedback. In our original manuscript, we included more background knowledge in the introduction to highlight the motivation behind our work. Although other reviewers found the presentation clear, we appreciate the comment about organization.
> Regarding the methodology section, due to the page limit, we include the most important concepts in the main paper. The diagram for the mask autoregressive sampling framework is provided in Appendix D as described in the original caption of Figure 2. The model architecture of the energy-scoring module is elaborated in Appendix E. To better guide the reader, we have updated the caption of Figure 2 to explicitly reference Appendix E in the updated version of our manuscript.
>
> Regarding the correspondence between equations and model components, we have annotated the important terms in Figure 2 in the original manuscript.
>
> For example, $\mathcal{L}_\text{energy}$ and $\mathcal{L}_\text{distill}$
> annotated in Figure 2 correspond to Equations (3) and (5). We avoid putting the entire equations in the figure for simplicity.
>
> **Q: The dataset processing description is confusing. In particular, the approach to handling audio longer than 10 seconds and the captioning process for AudioSet (which contains only label data) are not well explained.**
>
> Ans: We follow the standard preprocessing used in the literature [H][J][K] for text-to-audio generation. Specifically, we adopt IMPACT’s data preprocessing method exactly to ensure a fair comparison. For audio longer than 10 seconds, we apply the same truncation strategy as IMPACT and retain the original captions, which is also common practice in prior works that segment audio without modifying the associated text [J][K]. We used the same caption, despite truncating audio that was longer than 10 seconds.
> AudioSet does not have captions originally, so we follow AudioLDM2’s [K] method by using CLAP audio embeddings instead. There are no FlanT5 embeddings since FlanT5 requires text as input. We duplicate the CLAP audio embeddings to match the maximum configured text embedding length.
>
> [H] Huang et al. "IMPACT: Iterative Mask-based Parallel Decoding for Text-to-Audio Generation with Diffusion Modeling.", ICML 2025
>
> [J] Liu et al. "AudioLDM: Text-to-Audio Generation with Latent Diffusion Models.", ICML 2023
>
> [K] Liu et al. "Audioldm 2: Learning holistic audio generation with self-supervised pretraining.", TASLP 2024
>
>
> **Q: The subjective quality of the generated audio in the demos does not align with the quantitative results reported in the tables, and the improvements over state-of-the-art consistency-based models are not convincing.**
>
> Ans: Please note that even the ground truth only received an average MOS score of 3.68 on subjective audio quality (OVL). In IMPACT’s original paper (https://arxiv.org/abs/2506.00736), the ground truth only received an average MOS score of 3.57 on audio quality. This is reasonable since many ground truth samples were real-world recordings with wind blowing through the microphone, engine sounds of low quality, noises of crowds, … etc. Thus, it is normal for MOS scores of all models receiving scores less than 3.68, as they are bounded by the ground truth audio quality.
> We admit AudioDEAR’s sound quality falls behind its multistep sampling teacher, IMPACT, in the demo. However, the MOS scores we reported for AudioDEAR and IMPACT are 3.27 and 3.47, respectively. Hence, we argue that the audio quality in the demos aligns with the quantitative results reported in the OVL metric.
> Our method demonstrates consistent improvement on all metrics, including FD, FAD, KL, IS, CLAP, REL, OVL compared to consistency-based competitors (AudioLCM, ConsistencyTTA) with 1-step generation as shown in the Table below. Lastly, we would also like to highlight that IMPACT with 1 diffusion sampling step (Table 4, Diffusion) is unable to generate any intelligible audio.
>
> | model (1 step)   |   FD  |  FAD  |  KL  |   IS  |  CLAP |
> |------------------|-------|-------|------|-------|-------|
> | ConsistencyTTA   | 22.21 |  2.83 | 1.32 |  8.92 | 0.328 |
> | AudioLCM         | 25.36 |  4.44 | 1.74 |  8.25 | 0.267 |
> | AudioDEAR        | 18.67 |  2.79 | 1.06 |  9.66 | 0.334 |

---

> > ### Comment · Reviewer_Rb5X · 2025-11-25
> >
> > Thank you for the author’s response.
> > In your reply to the second question you stated that “There are no FlanT5 embeddings since FlanT5 requires text as input.”
> > But in the paper—specifically in Section 4.2 and Appendix C—you clearly describe that FlanT5 embeddings are used. This contradiction makes it very unclear how the model is actually implemented, what do you mean duplicate the CLAP audio embeddings, are you saying that when handling audioset data the system use CLAP audio embedding to replace the original Flat-T5 embedding?  Since the paper does not provide a concrete overall system architecture, it is difficult to understand how all components interact. At minimum, based on Section 4.2, it appears you are using both FlanT5 and CLAP, whereas your latest explanation claims that for the AudioSet model you only use CLAP. This suggests that there may be multiple system variants, but the paper does not clarify this.
> > Overall, many important details remain insufficiently explained. I will therefore keep my current score. Good luck with the submission.

---

> ### Author Response · Authors · 2025-11-25
>
> **The ablation study on sampling methods seems unconvincing, as existing flow-matching or consistency models should achieve comparable results with minimal sampling steps.**
>
> Ans: We assume the reviewer is referring to consistency models as Shortcut models and Meanflow, as in our original submission, there is no row named as consistency models in Table 4. In Shortcut model’s paper [L], they state that
> > shortcut model implementation there remains a gap between many-step generation quality and one-step generation quality
>
> in their limitation section. Thus, it is reasonable that Shortcut models cannot achieve comparable results with minimal sampling steps. For the Meanflow [M] training objective, though they demonstrate FID 3.43 with one-step sampling for the 676M model, it degrades to FID 6.17 for the 131M model, suggesting that model size matters. In Table 1 of [M], it is also shown that it is extremely sensitive to hyperparameters of guidance during training, with $w=1$ having FID 61.06 and $w=3$ having 15.53. Additionally, these results are all on the non-autoregressive DiT model, and are never shown that it should naturally work on masked autoregressive models.
>
> If the reviewer is referring to flow-matching models as described in work [N], these models learn the velocity field $x_1 - x_0$ from the inputs $(x_t, t)$ under the linear path $x_t = (1 - t) \cdot x_0+ t \cdot \varepsilon$.
> Below, we explain why this type of flow matching can not match diffusion models when only a few sampling steps are used. More specifically, they collapse when only one sampling step is allowed during inference.
>
> $v(x_t, t) = E[x_1 - x_0 | x_t, t]$ is the objective of the flow matching model that predicts the velocity $x_1 - x_0$ given $x_t$ and $t$. By doing one step directly from $t=1$ to $t=0$, the predicted clean version $\hat{x}_0$ is $x_1 - v(x_1, 1)$. With a perfectly trained model, $$\hat{x}_0 \approx x_1 -  E[x_1 - x_0 | x_1, 1] = x_1 - (x_1 - E[x_0|x_1]) = E[x_0|x_1]$$.
> This proves that a flow matching model with only one-step sampling degenerates to the mean of the training dataset.
>
> This phenomenon is also plotted in Figure 2 in [L]. Their high-level explanation is that at the very first step when $t=1$, inputs are pure noise and $(x_1, x_0)$ are randomly paired during training, so the learned velocity at $t=1$ points toward the dataset mean. This also explains why flow matching models are not able to achieve comparable results with minimal sampling steps. This phenomenon is termed as few-step ambiguity in [L]. In conclusion, flow matching learns the instantaneous velocity field, which typically results in curved trajectories in the data space. When a large step size is used to numerically solve the underlying Ordinary Differential Equation (ODE), the resulting discretization error is large, leading to poor-quality samples, such as mode collapse. Thus, we cast doubt on the claim that “flow matching models should achieve comparable results with minimal sampling steps”.
>
> [L] Frans, Kevin, et al. "One Step Diffusion via Shortcut Models." ICLR 2025.
>
> [M] Geng, Zhengyang, et al. "Mean flows for one-step generative modeling." NeurIPS 2025.
>
> [N] Liu, Xingchao, and Chengyue Gong. "Flow Straight and Fast: Learning to Generate and Transfer Data with Rectified Flow." ICLR
>
> **Q: How many participants were involved in the subjective evaluation on the 90 generated samples?**
>
> Ans: As stated in Section 4.3, each sample receives at least 9 independent ratings. More specifically, the total number of annotators involved in the subjective evaluation is 13.
>
> **Q: Which specific flow-matching approach or implementation was used for the experiments reported in Table 4?**
>
> Ans: The term “flow matching” in the manuscript refers to the flow matching method of [O][N]. No few-step strategies are adopted for the two rows of flow matching in Table 4. We will update the manuscript to avoid confusion on this. For implementation, we follow the implementation of [P] (https://github.com/OliverRensu/FlowAR/blob/main/models/flowloss.py#L59).
>
> [N] Liu, Xingchao, and Chengyue Gong. "Flow Straight and Fast: Learning to Generate and Transfer Data with Rectified Flow." ICLR
>
> [O] Lipman, Yaron, et al. "Flow Matching for Generative Modeling.", ICLR 2023
>
> [P] Ren, Sucheng, et al. "FlowAR: Scale-wise Autoregressive Image Generation Meets Flow Matching." , ICML 2025

---

> ### Author Response · Authors · 2025-11-28
>
> We thank the reviewer for the prompt response. To address your concerns, we clarify how the conditioning embeddings are processed as follows.
>
> Our training data involves AudioCaps, WavCaps, and AudioSet, with data preprocessing aligned with previous work in the literature [H][K]. During training, we further partition the data into two parts based on the caption availability.
>
> Case 1: Captions Available
>
> For datasets that include captions, specifically AudioCaps and WavCaps, we extract both CLAP text embeddings and Flan-T5 text embeddings following the preprocessing step described in [H]. Each CLAP text embedding has the shape (1, 512), while the Flan-T5 text embeddings have the shape (77, 1024) by default. To align their embedding dimensions, we repeat the CLAP text embedding along the embedding dimension to construct a vector with shape (1, 1024). We then concatenate these repeated CLAP embeddings with the Flan-T5 embeddings along the sequence length dimension, resulting in a combined text embedding sequence with the shape of (78, 1024) or $(L^\prime \times d^\prime)$ as shown in Figure 7 in the updated manuscript.
>
>
> Case 2: Captions Unavailable
>
> We follow the strategy in [K], where CLAP audio embeddings are used as conditioning inputs when captions are unavailable. For AudioSet, where the captions are not provided, we are unable to extract Flan-T5 text embeddings and CLAP text embedding for an audio. Instead, we use CLAP audio embeddings as the conditioning signal.  As in Case 1, for each audio, we repeat a CLAP audio embedding along the embedding dimension to obtain a vector with shape (1, 1024). After that, we then duplicate this vector 78 times along the sequence-length dimension to obtain an embedding sequence with the shape (78, 1024). By doing so, the shape of the resulting embedding sequence matches that of the text embedding in Case 1. We do this because we also need a conditioning embedding sequence that has the same shape as the text embeddings mentioned in case 1.
>
> Lastly, we want to clarify that our best reported result is obtained using a **single system**. We do **not** use multiple systems or architecture variants for different training data. We appreciate the reviewer’s suggestion on this point.
>
> We would also like to note that we included diagrams illustrating our model architecture and procedure in Figures 2, 4, and 5 and Appendix D and E in our original manuscript. To further assist the future readers, we have updated our manuscript and added two figures (Figures 7 and 8) in Appendix I, providing more detailed visualizations of our architecture and training/inference pipeline.
>
> [H] Huang et al. "IMPACT: Iterative Mask-based Parallel Decoding for Text-to-Audio Generation with Diffusion Modeling.", ICML 2025
>
> [K] Liu et al. "Audioldm 2: Learning holistic audio generation with self-supervised pretraining.", TASLP 2024

---

### Official Review · Reviewer_xw9n · 2025-10-25

**Soundness:** 3
**Presentation:** 3
**Contribution:** 2
**Rating:** 6
**Confidence:** 4

**Summary:**

This paper proposes AudioDEAR, a energy scoring masked auto-regressive text-to-audio generation model.
The authors show that AudioDEAR can significantly accelerate a diffusion MAR method like IMPACT by removing the iterative de-noising procedure of IMPACT's diffusion head.
A distillation process, which enhances the learning by aligning it with the IMPACT teacher model in the transformer hidden space, further strengthens the result.
On the AudioCaps dataset, AudioDEAR balances inference speed and generation quality.

**Strengths:**

The proposed idea is novel, and the method is sound. The ablation study on loss function weight, whether to freeze the transformer, and number of examples for energy distance is thorough.

**Weaknesses:**

In the introduction, the authors claim that they are the first to apply the energy-distance objective in TTA generation, enabling one-step
latent synthesis with low latency. Based on my understanding, this statement may be misleading -- while the diffusion head is replaced with a single-step energy distance head, the overall generation process is still iterative, with 64 decoding iterations. Hence, I would disagree that the proposed method is truly "one-step", and invite the authors to clarify. That said, I agree that with a small model size, the total computation to generate each example is quite low, on par with truly single-step methods, as reflected in Figure 3.

Similarly, Table 1 and Figure 1 also refers to AudioDEAR as "one-step". They should be clarified.

Minor -- Figure 1 would be clearer if the x axis is in log scale.

**Questions:**

- The authors mention in Section 5.2 that the setting “freeze” denotes that the transformer backbone is initialized from IMPACT and kept frozen during training, while only the lightweight energy-scoring head is optimized. For distillation runs for which the transformer and the diffusion head are NOT freezed, are they similarly initialized with IMPACT?

- The IMPACT teacher in this paper seems to use less non-captioned training data than the original IMPACT paper, which used 5500 hours if I remembered correctly. Is there a reason for this modification?

- CFG is performed within the transformer of AudioDEAR according to section 3.2. If I remembered correctly, this contrasts with the IMPACT paper's implementation, where CFG is performed within the diffusion head. How much difference does this change bring? Does the IMPACT teacher/baseline in this paper also use transformer CFG?

- The single-diffusion-step performance of AudioDEAR is quite impressive, balancing inference speed and quality. Although one key advantage of energy-distance training is that the teacher is optional, it is unclear whether energy-distance approach can outperform consistency distillation in a distillation setting. It would be too much to ask for an experimental comparison between these two approaches, but I would appreciate some discussions on this matter.

---

> ### Author Response · Authors · 2025-11-25
>
> **Q: Based on my understanding, this statement may be misleading -- while the diffusion head is replaced with a single-step energy distance head, the overall generation process is still iterative, with 64 decoding iterations. Hence, I would disagree that the proposed method is truly "one-step", and invite the authors to clarify.**
>
> Ans: We thank the reviewer for pointing out this ambiguity. The term “one-step” in our work specifically refers to requiring just one step for the sampling module, not the whole autoregressive framework. We clarified this in the updated manuscript in footnote 1.
>
> **Q: The authors mention in Section 5.2 that the setting “freeze” denotes that the transformer backbone is initialized from IMPACT and kept frozen during training, while only the lightweight energy-scoring head is optimized. For distillation runs for which the transformer and the diffusion head are NOT freezed, are they similarly initialized with IMPACT?**
>
> Ans: Yes, they are also initialized with IMPACT.
>
> **Q: The IMPACT teacher in this paper seems to use less non-captioned training data than the original IMPACT paper, which used 5500 hours if I remembered correctly. Is there a reason for this modification?**
>
> Ans: We thank the reviewer for the positive feedback and for recognizing the novelty of our idea and the soundness of the method.
> The training data with 5500 hours is not an optimal setting as shown in IMPACT’s original paper. The 5500-hour set is mainly composed of audio from Audioset which does not provide text captions. In our work, we only adopted AudioCaps, WavCaps, and a 500-hour subset of Audioset as we found that this setup is enough to compete with baselines such as AudioLCM, ConsistencyTTA, and AudioTurbo. Using a small dataset configuration can also save training cost proportionally, making extensive ablation studies more feasible.

---

> ### Author Response · Authors · 2025-11-25
>
> **Q: CFG is performed within the transformer of AudioDEAR according to section 3.2. If I remembered correctly, this contrasts with the IMPACT paper's implementation, where CFG is performed within the diffusion head. How much difference does this change bring? Does the IMPACT teacher/baseline in this paper also use transformer CFG?**
>
> Ans: We thank the reviewer for raising the question regarding where classifier-free guidance (CFG) is applied in our model compared to IMPACT. We confirm that CFG  is applied to the output of the MLP diffusion sampling module (noise vectors) for IMPACT, and applied to the output of the transformer for AudioDEAR.
>
> **Theoretical clarification:**
>
> According to the original CFG paper [F], CFG is explicitly applied to the output noise vectors (also referred to as score estimates) of the diffusion model.
>
> Let $f(z, h)$ denote the sampling module of IMPACT, where $z$ is the input of the diffusion sampling module and $h$ is the transformer output serving as the condition.
>
> If $f$ were linear (or affine) in $h$, then applying CFG at the level of the conditioning vector $h$ would be mathematically equivalent to applying CFG at the level of the predicted noise vector $\epsilon$. Specifically, for linear $f$, we would have:
> $$f\left(z, h_{\text{uncond}} + w\cdot \big(h_{\text{cond}} - h_{\text{uncond}}\big)\right) = f(z, h_{\text{uncond}}) + w\cdot\Big( f(z, h_{\text{cond}}) - f(z, h_{\text{uncond}}) \Big),$$
> which shows that applying CFG before the sampling module is equivalent to applying CFG after the sampling module.
>
> **Why does this equivalence not hold for IMPACT?**
>
> In practice, however, the diffusion sampling head of IMPACT is not linear in $h$. It includes:
> - Adaptive Layer Normalization (Ada-LN), whose scale and shift parameters depend non-linearly on the conditioning
> - Non-linear activation functions within the MLP layers
>
> Because Ada-LN introduces multiplicative and non-affine transformations conditioned on $h$, the mapping $f(z,h)$ is non-linear in $h$. Consequently, $$f\left(z, (1-w)\cdot h_{\text{uncond}} + w\cdot h_{\text{cond}}\right) \neq (1-w)\cdot f(z, h_{\text{uncond}}) + w\cdot f(z, h_{\text{cond}}),$$ meaning that CFG applied to $h$ is no longer equivalent to CFG applied to the predicted noise $\epsilon$.
>
> **Why does AudioDEAR apply CFG at the transformer output level $h$?**
>
> Our AudioDEAR framework uses an energy-scoring head, which acts as an "implicit generative model". This means the network takes the output of the transformer $h$ and noise vector $n$ and directly outputs a sampled latent $x = f(n, h)$. Hence, the output is not a score or a probability distribution that one can easily apply CFG to.
>
> As a consequence, the only way to influence the generated distribution is to modify the parameters that the generated distribution is conditioned on, leaving applying CFG at the transformer output level a straightforward way to do so. As pointed out in the response to reviewer 1qHm, [G] also applies the same methodology for CFG.
>
> [F] Ho, Jonathan, and Tim Salimans. "Classifier-free diffusion guidance." arXiv preprint arXiv:2207.12598 (2022).
>
> [G] Efficient Speech Language Modeling via Energy Distance in Continuous Latent Space

---

> > ### Comment · Reviewer_xw9n · 2025-11-25
> > **Thank you for your response**
> >
> > I appreciate the authors for their response, and my questions have been mostly addressed.
> >
> > Regarding transformer CFG, it would be helpful to try it on the IMPACT teacher model and report the results, so that the effectiveness of different CFG settings can be disentangled from model performance. Do you think this is doable? Thank you.

---

> > > ### Author Response · Authors · 2025-11-27
> > >
> > > **Q: I appreciate the authors for their response, and my questions have been mostly addressed.
> > > Regarding transformer CFG, it would be helpful to try it on the IMPACT teacher model and report the results, so that the effectiveness of different CFG settings can be disentangled from model performance. Do you think this is doable?**
> > >
> > > Ans: We thank the reviewer for this insightful question. To address this, we conducted an ablation study where we replaced IMPACT’s original noise-prediction-level CFG with transformer CFG (i.e., representation-level CFG). The results are listed in Table F. Setting the CFG scale to 1.0 means there is no CFG applied, and hence, the results for both CFG strategies are the same. For transformer CFG, by setting CFG scales between 1.1 and 3.0, none of the variants in rows \(c) to (i) exceed the performance of IMPACT with original noise-prediction-level CFG with CFG scale 5.0 (row (a)).
> > > Rows (g)(h)(i) in Table F demonstrate severe collapse in generation quality when using a transformer CFG with CFG scale >= 3.0 (FD increases from 15.25 to 214.17 when the CFG scale is set to 5.0).
> > >
> > > We attribute this performance degradation of transformer CFG to the nonlinearity of the diffusion head and the error accumulation inherent in iterative diffusion sampling:
> > >
> > > 1. Non-linearity of the diffusion head:
> > >
> > > The diffusion head $F_\theta$ in IMPACT is a non-linear function (parameterized by MLPs with activation functions and Ada-LN conditions). Consequently, the linear interpolation of inputs does not result in a linear interpolation of outputs. Formally, for a CFG guidance scale $w$, and $z_t$ being the input of the diffusion head at diffusion step $t$:
> > > $$F_\theta(z_t, (1-w) \cdot h_{\text{uncond}} + w \cdot h_{\text{cond}}) \neq (1-w) \cdot F_\theta(z_t, h_{\text{uncond}}) + w \cdot F_\theta(z_t, h_{\text{cond}})$$
> > > Here, the Left-Hand Side (LHS) represents the output using transformer CFG ($\epsilon'$), and the Right-Hand Side (RHS) represents the standard noise-prediction-level CFG ($\epsilon$) used by IMPACT.
> > >
> > > 2. Error Accumulation in diffusion sampling:
> > >
> > > Unlike our proposed AudioDEAR model, which generates audio in a single step, IMPACT relies on a diffusion process that solves a differential equation over 100 sampling steps. At each step $t$, there is a discrepancy between the approximated direction $\epsilon'$ (derived from mixed representations with transformer CFG) and the true guided direction $\epsilon$. While this error might be manageable in a single-step sampling process, it accumulates over the 100-step trajectory. This accumulation causes the sampling process to diverge significantly from the target data manifold when the CFG scale is too large, resulting in the high FD and poor audio quality observed in Table F.
> > >
> > > This confirms that while transformer CFG is effective for our AudioDEAR model (where no trajectory accumulation occurs), it is suboptimal for multiple-step sampling diffusion baselines like IMPACT.
> > >
> > > Table F. IMPACT CFG ablation studies
> > >
> > > | IMPACT                           | CFG scale | FD     | FAD    | KL    | IS    | CLAP   |
> > > |----------------------------------|-----------|--------|--------|-------|-------|--------|
> > > | (a) noise-prediction-level CFG   | 5.0       | **15.25** | **1.26** | **1.06** | **10.57** | **0.372** |
> > > | (b) no CFG                       | 1.0       | 22.42  | 2.96   | 1.42  | 6.84  | 0.269  |
> > > | \(c) transformer CFG              | 1.1       | 21.00  | 2.55   | 1.34  | 7.07  | 0.282  |
> > > | (d) transformer CFG              | 1.5       | 17.12  | 1.93   | 1.20  | 8.37  | 0.313  |
> > > | (e) transformer CFG              | 2.0       | 16.13  | 1.70   | 1.18  | 9.16  | 0.327  |
> > > | (f) transformer CFG              | 2.5       | 16.94  | 1.69   | 1.23  | 8.92  | 0.321  |
> > > | (g) transformer CFG              | 3.0       | 26.73  | 3.08   | 1.65  | 8.46  | 0.269  |
> > > | (h) transformer CFG              | 4.0       | 115.96 | 20.71  | 1.65  | 8.46  | 0.055  |
> > > | (i) transformer CFG              | 5.0       | 214.17 | 49.22  | 4.51  | 1.17  | -0.042 |

---

> ### Author Response · Authors · 2025-11-25
>
> **Q: The single-diffusion-step performance of AudioDEAR is quite impressive, balancing inference speed and quality. Although one key advantage of energy-distance training is that the teacher is optional, it is unclear whether energy-distance approach can outperform consistency distillation in a distillation setting. It would be too much to ask for an experimental comparison between these two approaches, but I would appreciate some discussions on this matter.**
>
> Ans: Based on the results reported in the two papers, Shortcut [L] and Meanflow [M], consistency models and consistency distillation perform extremely poorly in one-step generation. They fall far behind both Shortcut models and Meanflow in sample quality.
> In the related work section of [L], they state that:
> > consistency training enforces consistency among empirical $x_t$ and $x_{t+d}$ samples, which accumulates irreducible bias due to ambiguity
>
> For any given $x_t$, the underlying clean sample is not unique. Many distinct clean samples could map to the same partially corrupted observation $x_t$. Consequently, the training target for the model is ambiguous. This requires the model to resolve conflicting supervision signals arising from multiple plausible clean samples that could have produced the observed noisy samples. The model therefore converges toward a compromise solution, an average over these possibilities, which induces a persistent bias that cannot be eliminated by further training. This irreducible ambiguity in the supervision signal is a fundamental limitation of consistency-based methods and directly contributes to their degraded performance, particularly in one-step generation settings where errors from such bias are amplified. A method that already struggles to produce accurate latents in a one-step non-autoregressive setting will typically perform even worse when its errors are recursively reused in an autoregressive decoding pipeline.
>
> Although representation distillation can improve the quality of the conditioning signal by aligning the student transformer’s representations with those of the teacher, it does not alter the intrinsic limitations of the sampling module. In particular, the fundamental ambiguity inherent in consistency-based supervision arising from the non-uniqueness of the clean sample corresponding to a partially noised observation remains unresolved. As a result, even under a representation distillation framework, consistency distillation continues to inherit this irreducible bias, which helps explain why it may still perform poorly in generative settings.
>
> [L] Frans, Kevin, et al. "One Step Diffusion via Shortcut Models." ICLR 2025.
>
> [M] Geng, Zhengyang, et al. "Mean flows for one-step generative modeling." NeurIPS 2025.

---

> ### Author Response · Authors · 2025-11-27
>
> Dear reviewer,
>
> We would like to draw your attention to our rebuttal, in which we have carefully addressed each of your comments and included additional experimental evidence. Should our clarifications satisfactorily resolve the issues you raised, we would greatly appreciate your consideration in revising the score. Thank you for your time and thoughtful review.

---

> > ### Comment · Reviewer_xw9n · 2025-11-27
> > **Thank you for the reponse**
> >
> > I appreciate the authors' ablation study on representation CFG vs diffusion CFG.
> > This is super helpful insight for models with an AR/MAR+diffusion hybrid architecture.
> > Well done!

---

> > > ### Author Response · Authors · 2025-11-27
> > >
> > > We thank the reviewer for their appreciation of our response. If concerns are addressed, would you be willing to uprate your score?
> > >
> > > Thank you.

---

### Official Review · Reviewer_8LwK · 2025-10-27

**Soundness:** 2
**Presentation:** 2
**Contribution:** 2
**Rating:** 4
**Confidence:** 3

**Summary:**

The paper proposes a one-step text-to-audio (TTA) generation framework that replaces multi-step diffusion with energy-distance training and representation-level distillation from a diffusion-based teacher. This approach achieves up to 25× faster inference than state-of-the-art models (IMPACT).

**Strengths:**

**1. Effective Integration of Energy-Distance Loss and Distillation**

The paper successfully combines energy-distance loss with representation-level distillation from a diffusion teacher, enabling one-step text-to-audio generation while maintaining strong alignment between audio fidelity and text semantics.

**2. Significant Improvement in Inference Speed**

The proposed model achieves over 25× faster inference compared to diffusion-based baselines (e.g., IMPACT), demonstrating strong efficiency.

**Weaknesses:**

**1. Performance Gap with Teacher Model**

Unlike AudioLCM (Make-An-Audio as teacher), the proposed method shows a noticeable performance degradation compared to its diffusion-based teacher model (IMPACT), indicating that the distillation and energy-distance training did not fully preserve the teacher’s capability.

**2. Limited Inference Advantage in Small Batches**

Although the model achieves large speedups overall, its inference latency is slower than AudioLCM when batch size = 1 or 2, suggesting that the efficiency gain primarily appears in large-batch scenarios and may not generalize well to real-time or low-latency applications.

**Questions:**

**1. On Distillation and Teacher Model Fidelity**

Do the authors believe that the proposed distillation and energy-distance loss can fully replicate the teacher model’s performance? If not, what aspects of the teacher are hardest to transfer through this training scheme?

**2. On Inference Efficiency in Small-Batch Settings**

The proposed model shows slower inference than AudioLCM when batch size is 1–2. Is there a way to further optimize for improving small-batch efficiency? Alternatively, is this limitation inherent to the IMPACT-based autoregressive backbone used in the model?

---

> ### Author Response · Authors · 2025-11-25
>
> **Q: Unlike AudioLCM (Make-An-Audio as teacher), the proposed method shows a noticeable performance degradation compared to its diffusion-based teacher model (IMPACT), indicating that the distillation and energy-distance training did not fully preserve the teacher’s capability.**
>
> Ans: We assume the reviewer is referring to the two-step AudioLCM model, which is the variant reported to perform relatively close to its teacher, Make-An-Audio 2. However, according to Table 1 in the AudioLCM paper, even this two-step model still shows degradation on FAD, KL, and CLAP score. More importantly, our work focuses specifically on one-step generation. When comparing AudioLCM with its teacher under that setting, the gap becomes much clearer: the one-step AudioLCM model is substantially worse than its teacher, Make-An-Audio 2. (Please also note that Make-an-audio and Make-an-audio 2 are different models.)
>
> | Model                                       | Step |   FD   |  FAD  |  KL  |   IS  |  CLAP  |
> |---------------------------------------------|------|--------|-------|------|-------|--------|
> | Make-an-audio 2 (AudioLCM’s teacher)        | 100  | 16.23  | 2.03  | 1.29 | 9.95  | 0.345  |
> | AudioLCM (1 step)                           | 1    | 25.36  | 4.44  | 1.74 | 8.25  | 0.267  |
> | AudioLCM (2 step)                           | 2    | 20.01  | 2.17  | 1.48 | 9.89  | 0.308  |
> | IMPACT                                      | 100  | 15.25  | 1.26  | 1.06 | 10.57 | 0.372  |
> | AudioDEAR                                   | 1    | 18.67  | 2.79  | 1.06 | 9.66  | 0.334  |
>
> **Q: Although the model achieves large speedups overall, its inference latency is slower than AudioLCM when batch size = 1 or 2, suggesting that the efficiency gain primarily appears in large-batch scenarios and may not generalize well to real-time or low-latency applications.**
>
> Ans: We acknowledge that our method shows relatively lower efficiency than the baseline models at small batch sizes (1 or 2). However, since inference is typically performed in batched mode in cloud-serving scenarios, this limitation is unlikely to affect most practical applications. Latency of generating 10-second audio with batch size 1 or 2 for AudioLCM, ConsistencyTTA, and AudioDEAR is all less than one second (real time factor << 1). This should not cause problems for real-time or low-latency applications.
>
> **Q: Do the authors believe that the proposed distillation and energy-distance loss can fully replicate the teacher model’s performance? If not, what aspects of the teacher are hardest to transfer through this training scheme?**
>
> Ans: As shown in Table 1 in our paper, distilling representations yields a substantial performance gain across all metrics. Specifically, FD, FAD, KL (where lower values are better) decrease by 3.42 points (≈15.5%), 1.03 points (≈27.0%), and 0.16 points (≈13.1%), while IS and CLAP (where higher values are better) increase by 1.59 points (≈19.7%) and 0.036 (≈12.1%). This clearly demonstrates the effectiveness of teacher-guided representation distillation.
>
> We restrict distillation to the transformer output, as the MLP sampling module in the teacher and student models are not functionally aligned. The teacher’s MLP sampling module is responsible for noise prediction with multiple sampling steps (e.g., n=100), while the student’s MLP sampling module is designed to predict samples (latents) in a one-step sampling (e.g., n=1) process.
> Because our model (n=1) is restricted to a single sampling step, a certain level of performance gap relative to the teacher (n=100) is to be expected.
>
> Therefore, we respectfully clarify that the performance gap between AudioDEAR and the teacher model should not be interpreted as a failure to transfer certain capabilities. Instead, it reflects the fact that AudioDEAR operates under a one-step sampling budget and thus cannot produce a smooth ODE trajectory like the teacher.

---

> ### Author Response · Authors · 2025-11-25
>
> **Q: The proposed model shows slower inference than AudioLCM when batch size is 1–2. Is there a way to further optimize for improving small-batch efficiency? Alternatively, is this limitation inherent to the IMPACT-based autoregressive backbone used in the model?**
>
> Ans: We thank the reviewer for the insightful comments on efficiency. An intuitive way to improve the efficiency is to reduce the autoregressive (AR) steps, however, it may reduce the model performance as the model needs to handle more tokens in a single step. We ran an extra experiment in different AR steps as shown in Table E by reducing the AR steps in AudioDEAR from 64 to 32. This change decreases the inference time from 0.79 s to 0.41 s, which is faster than the 0.61 s reported for AudioLCM, while still achieving better performance on three of the five objective metrics (FD, KL, and CLAP).
>
> Table E. Comparison of objective performances and inference time between AudioLCM and AudioDEAR.
> | model      | AR steps \(r) | sampling steps (n) |   FD   |  FAD  |  KL  |   IS  | CLAP  | latency (s) |
> |------------|--------------|--------------------|--------|-------|------|-------|-------|-------------|
> | AudioLCM   | --           | 2                  | 20.01  | 2.17  | 1.48 | 9.89  | 0.308 | 0.61        |
> | AudioDEAR  | 32           | 1                  | **19.16** | 3.01  | **1.08** | 9.39  | **0.328** | 0.41        |
> | AudioDEAR  | 64           | 1                  | **18.67** | 2.79  | **1.06** | 9.66  | **0.334** | 0.79        |

---

> ### Author Response · Authors · 2025-11-28
>
> Dear reviewer,
>
> We would like to draw your attention to our rebuttal, where we have carefully responded to each of your comments and included further experimental evidence. Should our clarifications satisfactorily resolve the issues you raised, we would greatly appreciate your consideration in revising the score. Thank you for your time and thoughtful review.

---

### Official Review · Reviewer_1qHm · 2025-11-01

**Soundness:** 3
**Presentation:** 4
**Contribution:** 2
**Rating:** 4
**Confidence:** 5

**Summary:**

This paper proposes replacing the diffusion head in IMPACT, which is a masked autoregressive diffusion-based text-to-audio model, with a one-step MLP generator trained by minimizing energy distance. The proposed AudioDEAR model is initialized with IMPACT, and then fine-tuned with energy distance loss and a feature matching loss between the student AudioDEAR model and the teacher IMPACT model.

**Strengths:**

The paper is well written, and the theoretical derivations are easy to understand. The proposed method is novel, at least for text-to-audio models.

The authors included many comparison results, including comparisons with MeanFlow and Shortcut. Although including more diffusion distillation would make it better. For example, variational score distillation methods such as DMD and adversarial distillation methods (GAN).

**Weaknesses:**

The audio quality sounds much worse compared to the demo audios from the IMPACT website. You should also provide a comparison between the IMPACT demo and your reproduction on the demo page.

Conducting CFG as described in Section 3.2 lacks theoretical grounds. Interpolating encoder output does not provide any guarantee on the sampled distribution. I saw [1] also introduced a different CFG variant for one-step generators. How well does it work compared to the proposed method?

MAR [2] reported that "during inference, the diffusion sampler has a decent cost of about 10% overall running time." So this raises concerns about whether eliminating the diffusion sampling from IMPACT really matters that much.

All comparisons between IMPACT and AudioDEAR are evaluated with 64 encoder steps and 100 diffusion steps for IMPACT. Is the trade-off between latency and sample quality still appealing with fewer encoder/diffusion sampling steps in IMPACT? Can you provide more analysis of the inference FLOPs as well?

[1] Efficient Speech Language Modeling via Energy Distance in Continuous Latent Space, URL: https://arxiv.org/abs/2505.13181

[2] Autoregressive Image Generation without Vector Quantization, URL: https://arxiv.org/abs/2406.11838

[3] https://audio-impact.github.io/

**Questions:**

When it comes to inference latency, engineering details matter. What kind of implementation are you using for testing inference latency? Did you use technologies such as FlashAttention and CUDA Graphs for accelerating inference? Is the acceleration still significant on more recent NVIDIA GPU architectures? I only see V100 evaluation results in the paper.

What is the current status of research in accelerating specifically MAR-like models?

What do you think is the main cause of existing diffusion distillation methods that work on image generation not working on masked diffusion autoregressive models such as MAR?

---

> ### Author Response · Authors · 2025-11-25
>
> **Q: The audio quality sounds much worse compared to the demo audios from the IMPACT website. You should also provide a comparison between the IMPACT demo and your reproduction on the demo page.**
>
> Ans: We thank the reviewer for acknowledging that our proposed method is novel and that the paper is well-written. In the paper, we have demonstrated that adding representation distillation can significantly narrow down the gap between our proposed AudioDEAR model and the IMPACT teacher model. We admit that there is still a remaining gap in terms of objective performance, however, we also want to emphasize that our proposed model significantly improves the inference speed compared to IMPACT. While having low-latency inference, AudioDEAR also exceeds the performance of AudioLCM, ConsistencyTTA, and AudioTurbo on several objective and subjective metrics. In the demo, it is also clear that our model performs better than these existing baseline models. In conclusion, we emphasize that the strength of AudioDEAR isn’t in matching the teacher’s performance exactly, but in being much faster and outperforming many other existing audio generation models.
> Following the reviewer's instructions to compare the official IMPACT demo with our model, we have created a new demo page and uploaded it to the supplementary materials.
>
> **Q: Conducting CFG as described in Section 3.2 lacks theoretical grounds. Interpolating encoder output does not provide any guarantee on the sampled distribution. I saw [A] also introduced a different CFG variant for one-step generators. How well does it work compared to the proposed method?**
>
> Ans: Thank you for pointing this out. We agree with the reviewer that our CFG methodology does not directly correspond to its classical definition (in the sense of mixing conditional probability distributions), however our approach has been successfully leveraged in prior work. As the Reviewer has mentioned in [A], Eq. (12) is exactly the same as the one in Eq. (4) of our paper. Both extrapolate the output of the conditional vector before forwarding to the sampling module.
> In our paper, AudioDEAR mixes the conditional and unconditional transformer representations before the one-step energy head:
> $$h^i = \mathrm{CFG} \cdot h^i_{\text{cond}} + (1 - \mathrm{CFG}) \cdot h^i_{\text{uncond}}$$
> This is a linear extrapolation or extrapolation between $h^i_{\text{uncond}}$ and $h^i_{\text{cond}}$. Rewriting gives:
> $$h^i = h^i_{\text{uncond}} + \mathrm{CFG} \cdot (h^i_{\text{cond}} - h^i_{\text{uncond}})$$
> Eq. (12) of [A]:
> SLED applies the same operation to its autoregressive representation $z_t$ before the per-token generative MLP:
> $$z_t^{\text{cfg}} = z_t' + \lambda \cdot (z_t - z_t')$$,
> where $z_t$ is the conditional representation and $z_t'$ is the unconditional one.
> If we set $\lambda = \mathrm{CFG}$ and identify $z_t' \leftrightarrow h^i_{\text{uncond}}, \quad z_t \leftrightarrow h^i_{\text{cond}}$, then the two equations are algebraically identical. Both implement CFG by moving from the unconditional hidden state toward the conditional one before the sampling head. They are the same operation for CFG written just with different symbols, and both are applied at the model’s hidden-state level. We have cited [A] in our updated manuscript in Section 4.2 near Eq. (4).
>
> [A] Efficient Speech Language Modeling via Energy Distance in Continuous Latent Space

---

> ### Author Response · Authors · 2025-11-25
>
> **Q: MAR [B] reported that "during inference, the diffusion sampler has a decent cost of about 10% overall running time." So this raises concerns about whether eliminating the diffusion sampling from IMPACT really matters that much.**
>
> Ans: We thank the reviewer for raising this point regarding the cost of the diffusion sampler. After carefully examining the official MAR [B] implementation, we believe the statement in the paper regarding the diffusion sampler’s 10% inference cost is based on the count of floating-point operations (FLOPs), not wall-clock inference latency. However, lower cost of FLOPs does not imply lower wall-clock inference latency [C][D].
> We show the resulting breakdown of FLOPs and inference latency in Table A, using the official implementation (https://github.com/LTH14/mar) and running on an NVIDIA A100 GPU with batch size = 1, we observe that the 100-step MLP diffusion sampler accounts for approximately 90% of the end-to-end inference time and 10% of the FLOPs with MAR Large. For reference, we provide the scripts we used for FLOPs and latency measurement:
>
> FLOPs: https://anonymous.4open.science/r/mar-flop-latency-455E/cal_flops.py
>
> Latency: https://anonymous.4open.science/r/mar-flop-latency-455E/cal_latency.py
>
> As the experiments show, even though the transformer in MAR has most of the model’s parameters, it is not the main source of slow inference time. Instead, the diffusion sampler takes most of the runtime. This happens because the diffusion sampler must run 100 MLP steps in sequence, where each step depends on the previous one. These steps cannot be parallelized, no matter how strong the GPU is. So even though each MLP step has lower FLOPs, the overall process is still slow because it must run step-by-step.
>
> [B] Li et al., Autoregressive image generation without vector quantization. NeurIPS 2024
>
> [C] Fernandez et al.,The framework tax: Disparities between inference efficiency in nlp research and deployment. EMNLP 2023.
>
> [D] Chen et al.,. Run, don't walk: chasing higher FLOPS for faster neural networks. CVPR 2023
>
> Table A. Efficiency metrics of the wall-clock inference time and floating-point operations (FLOPs) for the MAR models in [B] with AR steps (r = 64), sampling steps (n = 100) and enabling classifier-free guidance (CFG) on NVIDIA A100 with batch size = 1.
>
>
> | Model     | Transformer (s) | MLP diffusion sampler (s) | MLP diffusion sampler (%) | Total (s) | Transformer FLOPs | MLP diffusion sampler FLOPs | MLP diffusion sampler FLOPs (%) | Total FLOPs |
> |-----------|------------------|----------------------------|-----------------------------|-----------|--------------------|-------------------------------|----------------------------------|--------------|
> | **MAR Large** | 1.95 | 13.63 | 87.5% | 15.58 | 1.98e+13 | 2.07e+12 | 9.5% | 2.19e+13 |
> | **MAR Base**  | 1.02 | 13.51 | 93.0% | 14.53 | 8.37e+12 | 2.05e+12 | 19.7% | 1.04e+13 |

---

> ### Author Response · Authors · 2025-11-25
>
> **Q: All comparisons between IMPACT and AudioDEAR are evaluated with 64 encoder steps and 100 diffusion steps for IMPACT. Is the trade-off between latency and sample quality still appealing with fewer decoding iterations / diffusion sampling steps in IMPACT?**
>
> Ans: To assess the effect of AR decoding steps (r: referred to as encoder steps by the reviewer) and sampling steps (n: referred to as diffusion steps by the reviewer) on IMPACT, we compare 6 configurations (IMPACT model a – f) in Table B. Across all settings, reducing either r or n consistently degrades the objective metrics.
>
> With r = 64, comparing IMPACT models (a) and (e) highlights this trade-off clearly: model (e) achieves very low latency by using only one sampling step (n=1), but the overall objective performance collapses. Notably, AudioDEAR also uses only one sampling step yet maintains competitive objective metrics, underscoring IMPACT’s limitations under this one-step sampling configuration.
>
> When r=4, IMPACT models (b), (c), and (d) show progressively worse objective performance as the sampling steps (r) decrease. Although IMPACT model (d) achieves latency comparable to our AudioDEAR model, its performance on objective metrics remains suboptimal.
>
> Overall, simply adjusting the number of AR decoding steps or sampling steps (models a – f) is insufficient for IMPACT to approach the performance of our AudioDEAR model.
>
> Table B. Comparing AR decoding steps (r), sampling steps (n), and objective performance between IMPACT and our AudioDEAR model with a NVIDIA V100 GPU.
> | Model (Base)       | AR steps (r) | sampling steps (n) |   FD    |  FAD  |  KL  |  IS   |  CLAP  | latency (s) |   FLOPs    |
> |--------------------|-------------|---------------------|---------|-------|------|-------|--------|-------------|------------|
> | (a) IMPACT         | 64          | 100   | 15.25   | 1.26  | 1.06 | 10.57 | 0.372  | 20.24       | 1.11e+13   |
> | (b) IMPACT         | 4           | 100   | 19.93   | 3.49  | 1.15 | 8.65  | 0.325  | 1.21        | 2.64e+12   |
> | (c) IMPACT         | 4           | 75   | 21.60   | 3.63  | 1.27 | 8.37  | 0.313  | 1.10        | 2.12e+12   |
> | (d) IMPACT         | 4           | 50  | 36.30   | 7.63  | 1.73 | 6.51  | 0.230  | 0.62        | 1.60e+12   |
> | (e) IMPACT         | 64          | 1   | 128.47  | 34.44 | 4.94 | 1.18  | -0.047 | 0.83        | 9.00e+12   |
> | (f) IMPACT         | 6          | 50   | 24.54  | 3.83 | 1.47 | 7.70  | 0.273 |    1.20     |  1.88e+12  |
> | AudioDEAR (ours)   | 64          | 1    | 18.67   | 2.79  | 1.06 | 9.66  | 0.334  | 0.79        | 8.99e+12   |
>
>
> **Can you provide more analysis of the inference FLOPs as well?**
>
> Ans: We show the FLOPs for both IMPACT and our AudioDEAR under different AR steps and sampling steps. Compared to IMPACT (r = 64, n = 100), our AudioDEAR (r = 64, n = 1) model achieves 25x speed up on wall-clock inference time and 20% FLOPs saving.
>
> Table H. Comparing IMPACT model (a) and AudioDEAR.
> | Model (Base)       | AR steps (r) | sampling steps (n) |   FD    |  FAD  |  KL  |  IS   |  CLAP  | latency (s) |   FLOPs    |
> |--------------------|-------------|---------------------|---------|-------|------|-------|--------|-------------|------------|
> | (a) IMPACT         | 64          | 100   | 15.25   | 1.26  | 1.06 | 10.57 | 0.372  | 20.24       | 1.11e+13   |
> | (e) IMPACT         | 64          | 1  | 128.47  | 34.44 | 4.94 | 1.18  | -0.047 | 0.83        | 9.00e+12   |
> | AudioDEAR (ours)   | 64          | 1 | 18.67   | 2.79  | 1.06 | 9.66  | 0.334  | 0.79        | 8.99e+12   |
>
> **Q: When it comes to inference latency, engineering details matter. What kind of implementation are you using for testing inference latency? Did you use technologies such as FlashAttention and CUDA Graphs for accelerating inference? Is the acceleration still significant on more recent NVIDIA GPU architectures? I only see V100 evaluation results in the paper.**
>
> Ans: We thank the reviewer for noting these acceleration techniques. Our latency measurement follows standard practice: we warm up the model and measure forward-pass time using `torch.cuda.Event(enable_timing=True)` with synchronization, repeating 100 times and averaging the results. We did not use FlashAttention or CUDA Graphs. With a sequence length of 256, their impact is expected to be limited, and all models were tested under the same conditions, so relative comparisons remain valid.
> As suggested, we supplement our original NVIDIA V100 results with measurements on a more recent NVIDIA A100 GPU, as shown in Table C. These results show that the latency reduction of AudioDEAR over IMPACT remains significant across GPU generations (25x → 20x).
>
> Table C. The inference time (s) comparison between IMPACT and our AudioDEAR model with AR decoding steps (r = 64) and enabling CFG on NVIDIA V100 and A100 GPUs with batch size 1.
>
> | Model      | v100  | a100  |
> |------------|-------|-------|
> | IMPACT     | 20.24 | 15.23 |
> | AudioDEAR  | 0.79  | 0.76  |

---

> ### Author Response · Authors · 2025-11-25
>
> **What is the current status of research in accelerating specifically MAR-like models?**
>
> Ans: Though there are lots of consistency-based methods and recent new modeling approaches like Shortcut and Meanflow, to the best of our knowledge, these methods have not been explored on MAR-like models by other existing work. We are the first to replace the diffusion objective of a MAR-like model (IMPACT) with the Shortcut and Meanflow objective for ablation studies on the text-to-audio generation task.
>
> **Q: What do you think is the main cause of existing diffusion distillation methods that work on image generation not working on masked diffusion autoregressive models such as MAR?**
>
> Ans: We respectfully point out that this question has some ambiguity due to the fact that MAR is also an image generation model. Therefore, we assume that the claim “existing diffusion distillation methods that work on image generation” means that existing methods that reduce sampling steps, which work on non-autoregressive ones, do not work on autoregressive ones. We acknowledge that multiple causes may contribute to the poor objective performance of accelerating MAR-like models. We provide one possible reason to address this question.
> Existing methods that aim to reduce the number of sampling steps, or more generally, flow map methods such as Shortcut and Meanflow, perform well on non-autoregressive methods. However, as shown in Table 4 of our manuscript, they can not be easily applied to MAR-like models. We explain this phenomenon by introducing the background on few-step sampling and discuss the possible underlying cause as follows.
>
> For flow matching during inference, it is impossible to take an infinite number of small steps (i.e., infinitesimal steps) in the ODE to generate a sample. Instead, it requires following the ODE with a predefined finite number of steps (time-discretization). This introduces an error between the results of following the true ODE with infinitesimal steps and the results of following the ODE with a finite number of steps. This is known as the discretization error resulting from time-discretization.
>
> Flow map models enable few-step generation by learning the long jump of the ODE solution of flow matching models. In the context of few-step sampling, a small sampling module (e.g., the MLP module of an MAR-based frame work)  trained with flow mapping has a chance of introducing a high discretization error, especially under a one-step sampling constraint. Since autoregressive models require generating content based on previously generated context, if the early-generated latents are erroneous, the error propagates severely, causing significantly low generation quality.
>
> In our work, the energy-scoring module samples from the predictive distribution rather than learning an ODE or flow map. Though AudioDEAR is also an autoregressive framework, it is free from time-discretization errors because it does not approximate any ODE solution at all, making it less prone to error propagation, as latents generated in early autoregressive iterations may have better quality.
>
> In summary, this is the logic of our response:
> - Flow map models (Shortcut, Meanflow) rely on approximating long ODE jumps.
> - Few-step sampling introduces a large time-discretization gap.
> - In MAR-like autoregressive pipelines, early-step error introduced by the small sampling head severely propagates.
> - Energy scoring avoids all ODE-induced time-discretization errors.
> - Therefore, our AudioDEAR model, which applies energy-scoring on a MAR-like framework, has a better chance of performing well under one-step sampling constraints.
>
> Lastly, we acknowledge that it is highly challenging to verify whether autoregressiveness is the root cause for the poor performance of flow mapping methods (Meanflow and Shortcut) on MAR-like models.
> One may argue that providing ground truth latents for early autoregressive iterations and using the MAR model to generate the remaining positions can help investigate whether the poor performance originates from early generation errors. However, this solution may still not work because in each autoregressive iteration in MAR, we randomly select positions to generate, so the generated positions at each iteration do not follow chronological order. As a consequence, evaluating the remaining generated subsegments becomes infeasible.

---

> ### Author Response · Authors · 2025-11-28
>
> Dear reviewer,
>
> We would like to draw your attention to our rebuttal, where we have carefully responded to each of your comments and included further experimental evidence. Should our clarifications satisfactorily resolve the issues you raised, we would greatly appreciate your consideration in revising the score. Thank you for your time and thoughtful review.

---

### Author Response · Authors · 2025-11-30
**Outline of rebuttal summary**

**Summary of this work:**

To address the high latency of autoregressive diffusion models in text-to-audio generation, we introduce **AudioDEAR**, a one-step sampling framework that combines energy-scoring training with representation-level distillation. AudioDEAR maps Gaussian noise directly to audio latents via an energy-scoring head, eliminating costly iterative sampling required by diffusion models. This yields a **25x speed-up** over the state-of-the-art multi-step method, IMPACT. Empirical results on AudioCaps demonstrate that AudioDEAR  **outperforms prior few-step baselines, including AudioLCM, ConsistencyTTA, and AudioTurbo on multiple objective metrics**, demonstrating its efficiency in one-step generation while generating audio with high quality.

Note: For the AC’s convenience, the following paragraphs include links to the detailed rebuttal responses. If a link misdirects, simply scroll and refresh.

**Incorrect claims by reviewers:**

* Reviewer 1qHm’s incorrect claims about classifier-free guidance (CFG) ([our response](https://openreview.net/forum?id=QU8raq5UhG&noteId=QMdQJOOD0I))
* Reviewer 1qHm’s comments about MAR’s diffusion sampler efficiency cost (FLOPs vs. wall-clock inference time) misled by the original MAR paper ([our response](https://openreview.net/forum?id=QU8raq5UhG&noteId=lSDdPt8rUU))
* Reviewer Rb5X’s incorrect claims about flow matching and evidence on hallucinating on experiments that do not exist ([our response](https://openreview.net/forum?id=QU8raq5UhG&noteId=cVbxNaQdfe))

**Strengths Highlighted by Reviewers:**

* Good novelty and presentation (xw9n and 1qHm)

**Concerns or questions raised by reviewers requesting new experiments or updates:**

* FLOPs analyses  ([Table H](https://openreview.net/forum?id=QU8raq5UhG&noteId=zl2QeLgIbJ))
* Measuring inference latency with a newer version of GPUs  ([Table C](https://openreview.net/forum?id=QU8raq5UhG&noteId=zl2QeLgIbJ))
* IMPACT’s AR decoding iterations/diffusion steps tradeoff ([Table B](https://openreview.net/forum?id=QU8raq5UhG&noteId=zl2QeLgIbJ))
* Transformer-level CFG vs noise-prediction-level CFG ([Table F](https://openreview.net/forum?id=QU8raq5UhG&noteId=UPygqUhAfl))
* Implementation details regarding the text embeddings (not major focus of this work) (Appendix I)

**Concerns or questions raised by reviewers without the need for new experiments:**

* Performance gap between the teacher model (IMPACT) and AudioDEAR
* Potential reason for existing few-step sampling methods not working on MAR
* Model initialization clarification

**Rebuttal status:**

* Positive feedback from reviewer xw9n. All concerns addressed.
* Received followup requests from reviewer Rb5X. Followup requests fully addressed.
* No response from reviewers 1qHm and 8LwK throughout the whole duration of the rebuttal phase.

---

> ### Author Response · Authors · 2025-11-30
> **Incorrect claims by reviewers**
>
> **Reviewer 1qHm’s incorrect claims about classifier-free guidance (CFG) ([our response](https://openreview.net/forum?id=QU8raq5UhG&noteId=QMdQJOOD0I))**
>
> The reviewer incorrectly claimed that the CFG method in reference \[A\] differs from ours. Consequently, the reviewer cited \[A\] and requested a comparison, implicitly implying \[A\]’s method as a better alternative. This is factually incorrect. As demonstrated in our rebuttal, the two approaches are algebraically equivalent. Both our implementation and the method cited by the reviewer (Eq. 12 in \[A\]) perform the exact same linear extrapolation of hidden states prior to the sampling head.
>
> \[A\] Efficient Speech Language Modeling via Energy Distance in Continuous Latent Space
>
> **Reviewer 1qHm’s comments about MAR’s diffusion sampler efficiency cost (FLOPs vs. wall-clock inference time) misled by the original MAR paper ([our response](https://openreview.net/forum?id=QU8raq5UhG&noteId=lSDdPt8rUU))**
>
> The reviewer incorrectly claimed that the diffusion sampler in MAR \[B\] has a relatively low cost, with approximately 10% of the running time. Consequently, the reviewer questioned the practical significance of our method by reducing the cost of the sampler. However, our empirical reproduction confirms that this claim conflates theoretical FLOPs with actual wall-clock latency. While our experiments confirm that the sampler contributes only about 10% of the total FLOPs for the MAR large model, our profiling on an NVIDIA A100 shows that it nevertheless accounts for 87.5% of the total wall-clock inference time. This discrepancy exists because the diffusion sampler requires 100 sequential, non-parallelizable steps. In contrast, our method only requires one-step for sampling. Therefore, contrary to the reviewer’s assumption, replacing the diffusion sampler with our proposed energy-scoring method yields a substantial reduction in inference latency, validating the high practical significance of our approach.
>
> \[B\] Li et al., Autoregressive image generation without vector quantization. NeurIPS 2024
>
> **Reviewer Rb5X’s incorrect claims about flow matching and evidence on hallucinating on experiments that do not exist ([our response](https://openreview.net/forum?id=QU8raq5UhG&noteId=cVbxNaQdfe))**
>
> In our work, Table 4 ablates the MAR-like model with different few-step sampling methods. The reviewer incorrectly claimed that our Table 4 includes comparisons with "consistency models"; however, we only compare methods such as Shortcut \[L\] and Meanflow \[M\]. We did not inlcude consistency models in our submission, since \[L\] and \[M\] have already demonstrated that consistency models do not work well for one-step generation. Furthermore, the reviewer argued that flow matching models \[N\] should inherently achieve comparable results with minimal sampling steps. This is theoretically flawed. As established in \[L\], standard flow matching learns an instantaneous velocity field. When restricted to a single step (from $t=1$ to $t=0$), the prediction degenerates to the dataset mean ($E\[x\_0|x\_1\]$) due to "few-step ambiguity" and large time-discretization errors, leading to mode collapse.
>
> \[L\] Frans, Kevin, et al. "One Step Diffusion via Shortcut Models." ICLR 2025\.
>
> \[N\] Liu, Xingchao, and Chengyue Gong. "Flow Straight and Fast: Learning to Generate and Transfer Data with Rectified Flow." ICLR
>
> \[M\] Geng, Zhengyang, et al. "Mean flows for one-step generative modeling." NeurIPS 2025\.

---

> ### Author Response · Authors · 2025-11-30
> **Strengths Highlighted by Reviewers**
>
> **Good novelty and presentation (by reviewers xw9n and 1qHm)**
>
> We thank the reviewers xw9n and 1qHm for recognizing the **good novelty and presentation** of our work.

---

> ### Author Response · Authors · 2025-11-30
> **Concerns raised by reviewers requesting new experiments or updates**
>
> **FLOPs analyses (Requested by reviewer 1qHm) ([our response](https://openreview.net/forum?id=QU8raq5UhG&noteId=zl2QeLgIbJ))**
>
> As requested by reviewer 1qHm, we provided a detailed breakdown of floating point operations (FLOPs) in Table H to substantiate our efficiency claims. When comparing the standard high-quality IMPACT configuration ($r=64, n=100$) against our proposed AudioDEAR ($r=64, n=1$), our method achieves a 25x acceleration in wall-clock inference time (0.79s vs 20.24s) and a 20% reduction in total FLOPs ($8.99 \\times 10^{12}$ vs $1.11 \\times 10^{13}$).
>
> **Measuring inference latency with a newer version of GPUs (Requested by reviewer 1qHm) ([our response](https://openreview.net/forum?id=QU8raq5UhG&noteId=zl2QeLgIbJ))**
>
> Regarding inference latency, reviewer 1qHm questioned whether our reported speedups would persist on more recent GPU architectures. We conducted additional inference latency measurements (Table C) on an NVIDIA A100 GPU to supplement our original results measured by an NVIDIA V100 GPU. The data confirms that our efficiency gains are robust across hardware generations. Specifically, AudioDEAR achieves a \~20x speedup over IMPACT on the A100 GPU. Thus, addressing the reviewer's concern. This demonstrates that the substantial latency reduction is intrinsic to our one-step generation design rather than an artifact of specific hardware configurations.
>
> **IMPACT’s AR decoding iterations/diffusion steps tradeoff (Requested by reviewer 1qHm) ([our response](https://openreview.net/forum?id=QU8raq5UhG&noteId=zl2QeLgIbJ))**
>
> To address reviewer 1qHm’s inquiry regarding whether baseline models can achieve fast and high-quality generation, we conducted a comprehensive evaluation of the IMPACT baseline under various configurations of autoregressive (AR) steps ($r$) and diffusion sampling steps ($n$) in Table B. Our empirical results indicate that simply reducing $r$ or $n$ in the baseline model leads to a consistent and severe degradation in objective metrics, confirming that naively reducing the number of sampling steps is **not** a viable strategy for efficient generation.
>
> **Transformer-level CFG vs noise-prediction-level CFG (Requested by reviewer xw9n) ([our response](https://openreview.net/forum?id=QU8raq5UhG&noteId=UPygqUhAfl))**
>
> At the specific request of reviewer xw9n, we conducted an ablation study to disentangle the effectiveness of different CFG strategies from model performance. We applied "transformer-level CFG" (representation mixing) to the IMPACT teacher model and compared it directly against the standard "noise-prediction-level CFG". The results clearly demonstrate that while transformer-level CFG is effective for our single-step model, it is suboptimal for multi-step diffusion baselines like IMPACT. As detailed in our rebuttal ([Table F](https://openreview.net/forum?id=QU8raq5UhG&noteId=UPygqUhAfl)), replacing standard CFG with transformer-level CFG in IMPACT leads to catastrophic performance collapse at higher guidance scales (e.g., at scale 5.0, Fréchet Distance deteriorates from 15.25 to 214.17). We provided a theoretical justification for this behavior, attributing it to: (1) non-linearity of the diffusion head, and (2) error accumulation.
>
> Following this extensive ablation and theoretical clarification, Reviewer xw9n explicitly stated: **"I appreciate the authors for their response, and my questions have been mostly addressed."** We believe the reviewer was satisfied with the validity of our method, but was likely unable to raise their score to reflect this due to the OpenReview system lockdown in the middle of the discussion period.
>
> **Implementation details regarding the text embeddings (Questions and requests by reviewer Rb5X) ([our response](https://openreview.net/forum?id=QU8raq5UhG&noteId=vDetn0zgTf))**
>
> Reviewer Rb5X expressed confusion regarding our handling of text embeddings for datasets without captions (e.g., AudioSet), incorrectly perceiving a contradiction between our manuscript’s description of Flan-T5 usage and our rebuttal clarification. The reviewer mistakenly assumed that because FlanT5 embeddings cannot be generated for non-captioned data, our approach must rely on multiple disparate model variants. We clarified that our framework utilizes a single, unified system architecture regardless of the input data source. We achieve this by strictly following established preprocessing protocols from IMPACT and AudioLDM2. We have added detailed visualizations (Figures 7 and 8 in Appendix I) to resolve the reviewer’s misunderstanding of this pipeline. However, we respectfully note that these standard data preprocessing steps constitute auxiliary engineering details rather than the central scientific contribution of our work. The reviewer’s continued focus on these standard implementation practices, which merely ensure dimensional consistency, should not detract from the evaluation of our core contributions in one-step generative modeling.

---

> ### Author Response · Authors · 2025-11-30
> **Concerns raised by reviewers without the need for new experiments**
>
> **Performance gap between the teacher model (IMPACT) and AudioDEAR (Question of reviewer 1qHm and 8LwK)**
>
> ([Our response](https://openreview.net/forum?id=QU8raq5UhG&noteId=QMdQJOOD0I)) Reviewer 1qHm noted a performance gap between our proposed AudioDEAR model and its teacher, IMPACT, specifically pointing out that the audio quality does not strictly match the teacher's original demos. We acknowledge that, as with most one-step distillation methods, a gap in objective metrics remains. However, we emphasize that the primary contribution of AudioDEAR is achieving a massive reduction in inference latency while maintaining competitive quality. While we may not strictly match the teacher's 100-step performance, **AudioDEAR significantly outperforms other state-of-the-art one-step generation baselines, such as AudioLCM (26% decrease in FD) and ConsistencyTTA (16% decrease in FD), in multiple objective metrics and subjective listening tests.** To support this, we have provided a new demo page in the supplementary materials directly comparing our model against the official IMPACT demos as requested by reviewer 1qHm. From the demo, it is also very clear that AudioDEAR has better audio quality than two-step AudioLCM and one-step ConsistencyTTA.
>
> ([Our response](https://openreview.net/forum?id=QU8raq5UhG&noteId=92WHfi89bo)) Furthermore, reviewer 8LwK expressed concern that AudioDEAR seemed to suffer from greater degradation relative to its teacher than AudioLCM did relative to Make-An-Audio 2\. We clarified that this comparison was flawed, as the reviewer likely referred to the two-step variant of AudioLCM. When comparing the relevant one-step generation setting, AudioLCM exhibits severe degradation compared to its teacher (e.g., FD degrades 56%). In contrast, AudioDEAR preserves its teacher's capabilities much more effectively in the one-step regime (FD only degrades 22%). This confirms that our distillation and energy-distance training are highly effective for extreme one-step generation settings.
>
> **Potential reason for existing few-step sampling methods not working on MAR (Question of reviewer 1qHm) ([our response](https://openreview.net/forum?id=QU8raq5UhG&noteId=JWYsddxr64))**
>
> Regarding Reviewer 1qHm’s inquiry about why existing diffusion distillation (few-step sampling) methods fail on MAR-like models, we emphasize that this discussion focuses on other few-step sampling methods rather than challenging the validity and effectiveness of our method.
>
> The failure of existing few-step sampling methods on MAR-like models is because autoregressive generation amplifies the time-discretization errors inherent in approximating ODE trajectories. Existing flow-map methods (e.g., Shortcut \[L\], Meanflow \[M\]) rely on approximating ODE trajectories, where few-step sampling introduces unavoidable time-discretization errors. While non-autoregressive models can tolerate these errors, MAR-like models rely on early-generated tokens as context. Consequently, even minor time-discretization errors in early steps propagate and accumulate, severely degrading overall quality. Our proposed method avoids this problem by utilizing energy-scoring to sample directly from the predictive distribution, thereby eliminating ODE-induced time-discretization errors and ensuring good quality under one-step constraints.
>
> \[L\] Frans, Kevin, et al. "One Step Diffusion via Shortcut Models." ICLR 2025\.
>
> \[M\] Geng, Zhengyang, et al. "Mean flows for one-step generative modeling." NeurIPS 2025\.
>
>
> **Model initialization clarification (Question of reviewer xw9n) ([our response](https://openreview.net/forum?id=QU8raq5UhG&noteId=XtSFMOQhPs))**
>
> We confirm that the transformer backbone is initialized with pre-trained IMPACT weights for all distillation experiments, regardless of whether the backbone is subsequently frozen or fine-tuned.

---

> ### Author Response · Authors · 2025-11-30
> **Rebuttal status**
>
> **Reviewer xw9n**
>
> Authors posted the first response on 25 Nov 2025 at 23:25 (utc+8). The reviewer replied on 26 Nov 2025 at 03:12 (utc+8), saying “questions have been mostly addressed”, and followed up on requesting experiments of transformer CFG applied on IMPACT. The authors provided new experiments to fulfill this request on  27 Nov 2025 at 21:23 (utc+8), and got an appreciation of the reviewer saying the experiments are well done on 28 Nov 2025 at 04:26 (utc+8). Unfortunately, the discussion was cut off shortly after by the system.
>
> **Reviewer Rb5X**
>
> Authors posted the first response on 25 Nov 2025 at 23:40 (utc+8). The reviewer replied on 26 Nov 2025 at 00:21 (utc+8), asking for a more detailed overall structural diagram for our AudioDEAR framework. specifically targeting on the implementation details of text embedding extraction. The authors fulfilled the reviewer’s further request on 28 Nov 2025 at 18:33 (utc+8) by providing very detailed explanations and figures in the updated manuscript. Unfortunately, the discussion was cut off shortly after by the system. The authors had no chance to wait for the reviewer’s response.
>
> **Reviewers 1qHm**
>
> Authors posted the first response on 25 Nov 2025 at 23:04 (utc+8). The rebuttal fulfilled the request of a new demo page (updated in supplementary materials), clarified the incorrect claims of CFG from the reviewer, and provided a full computational investigation on models MAR, IMPACT, and AudioDEAR on the number of FLOPs. Unfortunately, the discussion was cut off before the reviewer could provide any responses.
>
> **Reviewer 8LwK**
>
> Authors posted the first response on 25 Nov 2025 at 23:23 (utc+8). The rebuttal fulfilled concerns about teacher-student performance gaps and small batch latency. Unfortunately, the discussion was cut off before the reviewer could provide any responses.
>
> **Summary of rebuttal status**
>
> In summary, most concerns are objective questions related to implementation, CFG, FLOPs, latency, and some ablations on the IMPACT baseline model. The authors believe the responses in the rebuttal, additional experiments, and revision of the manuscript have addressed the concerns fully. One subjective concern about demos is addressed by exactly following the instructions of the reviewer and provided in the supplementary materials.
>
> **We are confident that, if the discussion had not been cut short, the reviewers who did not have the opportunity to reply would have acknowledged that our rebuttal thoroughly addressed their requests, which might have led to higher overall ratings.**

---

> ### Author Response · Authors · 2025-11-30
> **Summary of PDF Revision**
>
> The bullet points below are provided to help readers follow the updates made to the manuscript.
>
> * **Line 107**: We added a footnote to address reviewer xw9n’s concerns about the terminology “one-step”.
>
> * **Lines 227-232**: We updated the caption of Figure 2 to include references to Appendix E and Appendix I,  providing more detailed structural diagrams to address reviewer Rb5X’s concern.
>
> * **Lines 486-496**: We include additional results in Table 6 that explore the tradeoff between latency and output quality in IMPACT, thereby addressing reviewer 1qHm’s inquiry regarding the “AR decoding iterations/diffusion steps tradeoff.”
>
> * **Pages 24 and 25**: In response to reviewer Rb5X, we have further elaborated on the overall structure of AudioDEAR. The newly included Figures 7 and 8 in Appendix I offer additional clarity on the framework and its components.

---

### Meta-Review · Area_Chair_NFaV · 2026-01-07

**Summary:**

A major concern shared by the reviewers is the clear quality gap between the proposed method and the teacher model. Other major concerns include the lack of theoretical grounding for the CFG implementation, limited advantages in small-batch inference settings, and insufficient analysis of inference speed for the baseline methods.

**Reviewer Concerns:**

The rebuttal provides extensive analysis and explanation of inference speed and clearly resolves the related concerns. Although a formal theoretical justification for the CFG implementation is still lacking, the rebuttal offers sufficient empirical support. However, the rebuttal does not directly address the performance gap with the teacher model, and the claim that small-batch inference is less common is neither persuasive nor supported by evidence. Given that the primary contribution of this work is improved inference speed, a stronger justification of its benefits in real-world inference settings is necessary.

**Reviewer Scores:**

Based on the rebuttal and the limited discussion to date, further discussion is unlikely to be productive unless the authors can provide additional experiments or analysis addressing the remaining concerns.

---

### Decision · Program_Chairs · 2026-01-26

Reject